# DNA polymerase α/primase extraction from chromatin by VCP/p97 restricts ATR activation during unperturbed DNA replication

Sara Rodríguez-Acebes [1,4], Rodrigo Martín-Rufo [2,4], Alicia Gómez-Moya[2], Scott B. Churcher[2], Alejandro Fernández-Llorente[2], Guillermo de la Vega-Barranco[2], Alejandra Perona[2], Pilar Oroz[2], Elena Martín-Doncel[3], Luis Ignacio Toledo[3], Juan Méndez [1] & Emilio Lecona [2] ✉

The replication stress response is an essential pathway that deals with the obstacles that halt the progression of DNA replication forks even during an unperturbed S phase. Basal activation of the ATR and CHK1 kinases prevents the premature firing of origins of replication during S phase, avoiding the activation of an excessive number of replication forks and the appearance of genomic instability. However, the mechanisms that regulate ATR activation in the unperturbed S phase have not been fully determined. Here we present evidence that the AAA ATPase VCP/p97 regulates the presence of the DNA polymerase α/Primase complex (POLA/PRIM) on chromatin, thus limiting its activity and hampering the subsequent activation of ATR by TOPBP1. As a consequence, inhibiting VCP/p97 activates ATR and CHK1 and leads to a cell cycle arrest in G2/M. We propose that the priming activity of POLA/PRIM in the lagging strand is one of the determinants of the basal activation of ATR during an unperturbed S phase and VCP/p97 limits this activation through the extraction of POLA/PRIM from chromatin.

DNA replication is the ordered sequence of events that leads to the faithful duplication of the genetic material in a cell while preventing the appearance of mutations and genomic alterations that fuel genomic instability. However, in each S phase the advance of replication forks is challenged by the presence of alterations in the DNA or difficult to replicate regions that lead to the stalling of DNA polymerases, the accumulation of single stranded DNA (ssDNA) that is protected through the binding of RPA, the activation of the replication stress response (RSR) and the replication-coupled DNA repair mechanisms[1–3].

The RSR is mainly driven by the ATR kinase and its downstream effector, the CHK1 kinase. Together, ATR and CHK1 stabilize stalled DNA replication forks, inhibit new origin firing and activate the G2/M checkpoint to prevent the progression into mitosis in the presence of unreplicated DNA[4]. Both kinases are essential, reflecting the fundamental role of the RSR in every round of DNA replication, even in the absence of exogenous DNA damage[5]. ATR activity fluctuates during S phase to control the rate of DNA replication, and it has been suggested that the basal activation of ATR either monitors the accumulation of

[1]DNA Replication Group, Molecular Oncology Programme, Spanish National Cancer Research Centre (CNIO), Madrid, Spain. [2]Chromatin, Cancer and the Ubiquitin System lab, Centro de Biología Molecular Severo Ochoa (CBM) CSIC-Universidad Autónoma de Madrid, Department of Genome Dynamics and Function, Madrid, Spain. [3]Center for Chromosome Stability, Institute for Cellular and Molecular Medicine, Faculty of Health and Medical Sciences, University of Copenhagen, Copenhagen, Denmark. [4]These authors contributed equally: Sara Rodríguez-Acebes, Rodrigo Martín-Rufo. ✉e-mail: elecona@cbm.csic.es

RPA or the number of active origins[6–8]. However, it is still unclear how the RSR controls unperturbed DNA replication.

DNA synthesis is preceded by the formation of pre-replicative complexes at origins in G1. Origin firing sets the start of S phase through the phosphorylation and activation of the CMG helicase, leading to the opening of the DNA double helix and the loading of the rest of the replication machinery[9]. During replication elongation the CMG helicase unwinds the parental DNA while the replicative DNA polymerases synthesize new DNA[10,11]. First, the DNA polymerase α/Primase complex (POLA/PRIM) generates RNA-DNA primers that are the substrate for DNA polymerase δ. In the leading strand, DNA polymerase δ is quickly replaced by DNA polymerase ε that travels with the CMG helicase to continuously synthesize this strand. In contrast, the lagging strand requires multiple priming events by POLA/PRIM to generate the discontinuous Okazaki fragments that are synthesized by DNA polymerase δ.

Ubiquitylation and SUMOylation of DNA replication factors influence every step in DNA replication and repair, constituting one of the key mechanisms to maintain genomic stability[12–14]. The proteomic analysis of nascent chromatin revealed the presence of a SUMO-rich environment associated to DNA replication forks[15,16]. We have shown that the ubiquitin/SUMO landscape in chromatin is maintained by the coordinated action of the deubiquitinase USP7 and the AAA ATPase VCP/p97[17,18]. While USP7 removes ubiquitin from SUMOylated proteins in the replication forks, VCP/p97 extracts SUMOylated and ubiquitylated proteins from chromatin during DNA replication. VCP/p97 is a molecular machine that assembles as a ring-shaped homohexameric complex with a central pore where the ATPase domains are located[19,20]. VCP/p97 extracts proteins from chromatin when they are no longer needed to allow their degradation or recycling. A wide variety of VCP/p97 cofactors mediate the binding of specific substrates through the recognition of ubiquitylated and SUMOylated proteins[20,21]. Recent biochemical data show that, once the ubiquitylated substrate has been positioned by the cofactors on top of VCP/p97, a distal ubiquitin molecule in the chain is unfolded to promote the subsequent extrusion of the protein through the central pore of the complex[22–24]. During DNA replication, VCP/p97 mediates the extraction and degradation of CDT1 to prevent re-replication in S phase[25,26], and the removal of the CMG complex and other replisome components upon DNA replication termination or in response to DNA damage[13,27–29].

Here we dissect the functions of VCP/p97 during DNA replication beyond the control of termination, and we identify POLA/PRIM as a target for this ATPase. We show that VCP/p97 limits the amount of POLA/PRIM on chromatin and its priming activity. The increased priming by POLA/PRIM upon VCP/p97 inhibition raises the basal level of activation of the RSR in S phase and activates the G2/M checkpoint. Our data suggest that VCP/p97 limits the activation of the RSR in an unperturbed S phase through the extraction of POLA/PRIM from chromatin contributing to the correct control of S phase progression and the maintenance of genomic stability.

## Results

### VCP/p97 controls DNA replication

To analyze the functions of VCP/p97 during DNA replication we synchronized HCT116 colon adenocarcinoma cells in G1/S phase using a single thymidine block. After allowing the progression of the cells into S phase for 2 h, the action of VCP/p97 was blocked using the specific allosteric inhibitor NMS873 (from here on VCPi)[30]. As we have reported before[18], the inhibition of VCP/p97 impairs DNA synthesis in a time- and dose-dependent manner as shown by the decrease in EdU incorporation measured by flow cytometry (Fig. 1A, B). To characterize this function in detail, we investigated the effects of VCP/p97 inhibition on the dynamics of DNA replication using stretched DNA fibers. Synchronized HCT116 cells were incubated sequentially with chlorodeoxyuridine (CldU) and iododeoxyuridine

(IdU) in the presence of VCPi to measure fork rate (Fig. 1C, FR) and origin firing (Fig. 1C, 1st label origins). Blocking the activity of VCP/p97 led to a slight decrease in fork speed (Fig. 1D) along with a strong inhibition of origin firing (Fig. 1E). The analysis of termination events did not reveal any differences (Supplementary Fig. 1A), as anticipated, because VCP/p97 is required for the disassembly of the replication machinery that only happens after the actual termination of DNA replication. We also measured the effect of VCP/p97 on fork symmetry using a modified version of the stretched DNA fiber assay to evaluate: (a) the progression of ongoing forks through the ratio of both labels (Fig. 1F, CldU/IdU); (b) origin asymmetry through the ratio between the two tracks for the second label in a fiber originating from a single origin (Fig. 1F, long/short)[31]. These parameters reflect fork stalling, either in ongoing forks (CldU/IdU ratio) or in one of the forks emanating from the same origin of replication (long/short ratio). The inhibition of VCP/p97 led to an increase in the CldU/IdU ratio (Fig. 1G), indicative of a high frequency of stalled forks rather than reduced fork speed/impaired replication processivity[31]. Consistently, VCPi also increased the degree of asymmetry between the two forks emanating from the same origin (Fig. 1H and Supplementary Fig. 1B-C). Furthermore, the percentage of "asymmetric origins" was increased 4-fold (Supplementary Fig. 1D).

We wondered whether the induction of fork stalling could be due to an effect of VCP/p97 on the RSR. G1/S-synchronized HCT116 cells were released for 2 h and then treated with hydroxyurea (HU), an agent that depletes the pool of dNTPs through the inhibition of the ribonucleotide reductase, leading to fork stalling and the activation of the RSR. As expected, the treatment with HU increased the phosphorylation of CHK1 by ATR (CHK1-S345P) and the levels of γH2AX (Fig. 1I, lanes 1–4). In addition, HU increased the phosphorylation of RPA2 (RPA2-S4/8 P), indicative of ssDNA accumulation and the activation of the DNA damage response (Fig. 1I, lanes 1–4). Inhibiting VCP/p97 did not change the activation of the RSR by HU, showing a similar pattern of CHK1 phosphorylation and γH2AX accumulation (Fig. 1I, compare lanes 2-4 with 5-7). We observed a reduction in the phosphorylation of RPA2 in the presence of VCPi after HU treatment (Fig. 1I, compare lanes 4 and 7). This effect is most likely due to the lower fork speed caused by VCP/p97 inhibition that could limit the amount of ssDNA generated in the presence of HU. Of note, the inhibition of VCP/p97 alone led to a slight but consistent increase in the phosphorylation of CHK1 without inducing changes in RPA2 phosphorylation (Fig. 1I, lanes 8–10). The quantification of 4 independent experiments revealed a $1.9 \pm 0.3$ increase in the CHK1-S345P/CHK1 ratio in VCPi-treated cells compared to control cells (Supplementary Fig. 1E). We conclude that the inhibition of VCP/p97 affects DNA replication and concomitantly activates ATR to a limited extent, i.e. without inducing fork collapse or DNA damage, suggesting a functional link between the effects on DNA replication dynamics and the activation of the RSR.

### A dual function of VCP/p97 in origin firing and DNA replication fork progression

The changes in origin firing and fork speed are usually inversely correlated to allow the complete copy of DNA in the duration of the S phase[32]. Interestingly, VCP/p97 inhibition induced a strong block in origin firing along with a small decrease in fork speed, suggesting that VCP/p97 is necessary for both processes. To confirm that VCP/p97 plays a role during elongation, synchronized HCT116 cells were released for 2 h into S phase before treating them for 2 additional hours with nucleosides, VCPi and an inhibitor of CDC7 (CDC7i), the kinase responsible for the activation of the CMG helicase[33]. As previously described[34,35], nucleoside supplementation (dNTPs) increased fork speed with a concomitant reduction in new origin firing (Fig. 2A, B). The fact that VCPi prevented the increase in fork speed induced by nucleosides without changing origin firing inhibition (Fig. 2A, B), supports a direct role for VCP/p97 in sustaining fork elongation. In

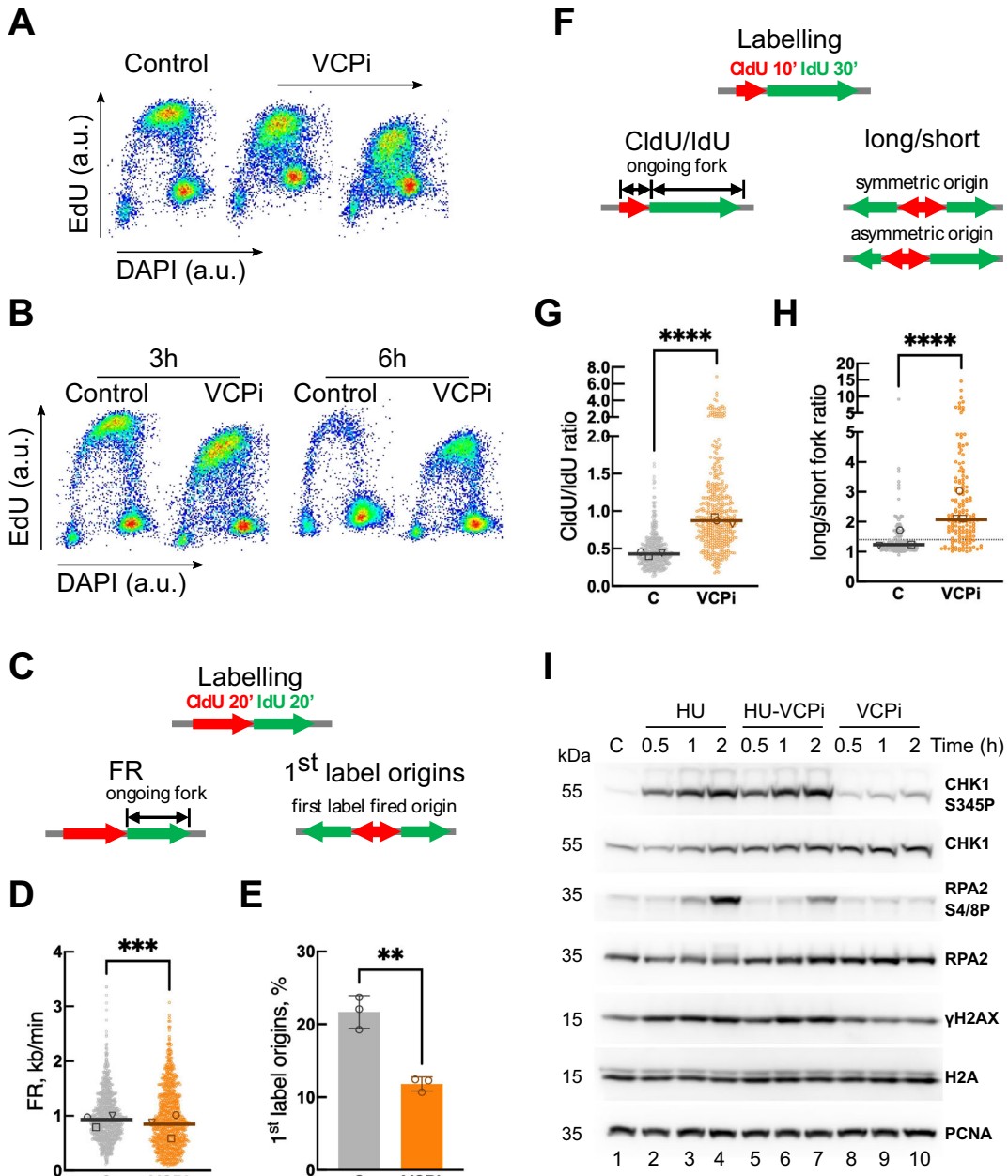

**Fig. 1 | VCP/p97 is necessary for DNA replication. A, B** Flow cytometry analysis of DNA content (DAPI) and DNA replication (EdU) in HCT116 cells, synchronized, released for 3 h and treated with 2 and 5 µM NMS873 (VCPi) for 3 h (**A**) or with 5 µM NMS873 (VCPi) for 3/6 h (**B**). One out of two representative experiments is shown. **C**–**H** DNA fiber analysis in HCT116 cells, synchronized, released for 2 h, treated for 2 h with 10 µM NMS873 (VCPi) or DMSO (control). **C** To measure fork rate (FR) and origin firing, cells were sequentially incubated with CldU and IdU for 20 min. FR was measured by the length of the IdU track. Origin firing (1st label origins) was measured as the percentage of CldU tracks flanked by IdU tracks. **D**, **E** show FR and 1st label origins upon treatment with 5 µM NMS873 (VCPi) for 2 h or DMSO (control, C). The pool (fork rate, bars represent the median of the pooled, ***$p = 0.0006$, Mann–Whitney test) or the average (1st label origins, %, mean ± SD, **$p = 0.0022$, $t$ test) of three experiments is shown. Symbols note the median of individual experiments. **F** For fork (CldU/IdU) and origin asymmetry (long/short), cells were sequentially incubated with CldU for 10 min and IdU for 30 min. The ratio of the track lengths CldU/IdU (CldU/IdU ratio) was used to follow fork stalling and the ratio of long and short IdU tracks arising from the same origin (long/short ratio) was used to follow origin asymmetry. **G** and **H** show the CldU/IdU ratio and long/short ratio upon treatment with 5 µM NMS873 (VCPi) for 2 h or DMSO (control, C). The pool of three independent experiments is shown. Bars represent the median of the pooled data. Symbols note the median of individual experiments. ****$p < 0.0001$ in (**G**, **H**), Mann–Whitney test. **I** Western blot analysis of whole cell extracts obtained from HCT116 cells, synchronized, released for 2 h and treated with 2 mM hydroxyurea (HU), 10 µM NMS873 (VCPi), a combination of both (HU-VCPi) or DMSO (lane 1, C). The indicated proteins were measured with specific antibodies. One of two repetitions is shown. Source data are provided as a Source Data file.

addition, inhibition of CDC7 led to a strong decrease in origin firing and a concomitant increase in fork rate (Fig. 2C-D). Similar to what we observed with nucleoside supplementation, the inhibition of VCP/p97 blocked the increase in fork rate induced by CDC7i without further inhibiting origin firing (Fig. 2C, D). Combined, these experiments show that VCP/p97 is required for fork elongation and suggest that it might

not be necessary for the activation of the signalling pathway that sets origin firing itself. In agreement with this notion, VCPi did not significantly affect the phosphorylation of MCM2 (Supplementary Fig. 2A).

We hypothesized that the apparent decrease in origin firing observed in DNA fibers after VCP/p97 inhibition (Figs. 1E, 2B, D)

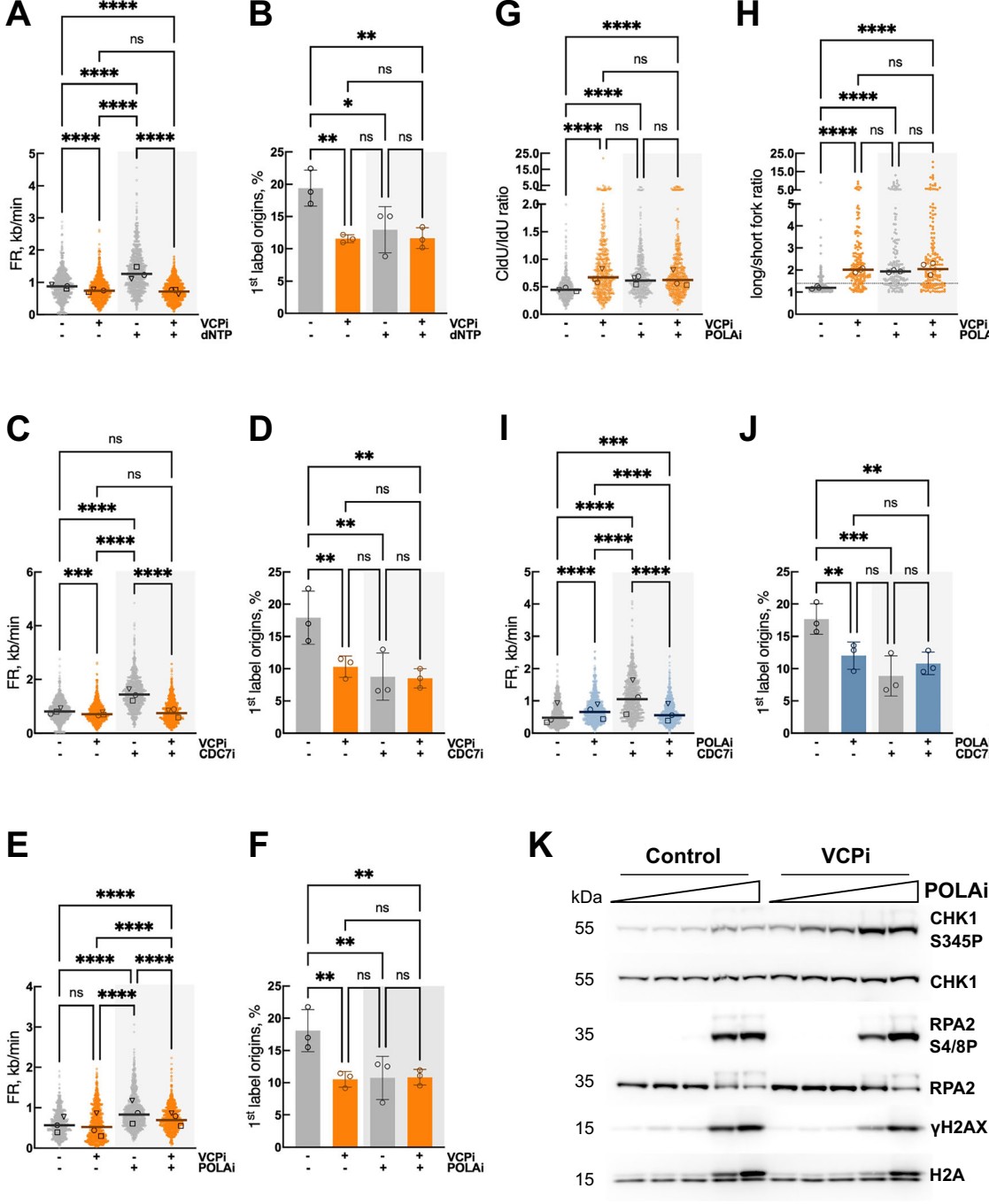

**Fig. 2 | Distinct effects of VCP/p97 in origin firing and fork progression.**
**A**–**F** DNA fiber analysis in HCT116 cells, synchronized, released for 2 h and treated for 2 h with: 10 μM VCPi, 150 μM dNTP, a combination of both or DMSO as a control (**A**, **B**); 10 μM VCPi, 20 μM CDC7i, a combination of both or DMSO (**C**, **D**); 10 μM VCPi, 0.5 μM POLAi, a combination of both or DMSO (**E**, **F**). FR (**A**, **C**, **E**) and 1st label origins (**B**, **D**, **F**) were determined. **G**, **H** DNA fiber analysis in cells synchronized, released for 2 h and treated for 2 h with: 10 μM VCPi, 0.5 μM POLAi, a combination of both or DMSO. CldU/IdU ratio (**G**) and long/short ratio (**H**) were determined. Individual data for the long/short ratio are shown in Supplementary Fig. 2B–E. **I**, **J** DNA fiber analysis in cells synchronized, released for 2 h and treated for 2 h with 0.5 μM POLAi, 20 μM CDC7i, a combination of both or DMSO. FR (**I**) and 1st label origins (**J**) were determined. For all DNA fiber analysis three independent experiments were performed; dot plots show pooled data and bars represent the median

(FR, CldU/IdU and long/short fork) and bar graphs (1st label origins) plot the mean ± SD. The median of individual experiments is noted with different symbols both in dot plots and bar graphs. *$p < 0.05$; **$p < 0.01$; ***$p < 0.001$; ****$p < 0.0001$; ns, non-significant, in Kruskall–Wallis with Dunn's post-test (**A**, **C**, **E**, **G**, **H**, **I**) or one-way ANOVA followed by Tukey's test (**B**, **D**, **F**, **J**). **K** Western blot analysis of whole cell extracts from HCT116 cells, synchronized, released for 2 h and treated for 2 h with increasing concentrations of POLAi (0.1, 0.2, 0.5, 1 μM) alone or in combination with 10 μM VCPi. The levels of total and phosphorylated CHK1 (S345), total and phosphorylated RPA2 (S4/S8), phosphorylated H2AX (γH2AX) and total histone H2A were measured with specific antibodies. The experiment was repeated three times and one representative result is shown. Source data are provided as a Source Data file.

could reflect a blockage of the steps in DNA replication that immediately follow origin activation. Using synchronized HCT116 cells released for 2 h into S phase, we blocked DNA replication right after origin firing by inhibiting DNA polymerase α with adarotene/ST1926 (POLAi)[36] for 2 additional hours. To understand how POLAi inhibition affects the priming activity of POLA/PRIM we directly measured the presence of primed DNA applying a modified version of the alkaline comet assay in combination with EdU labelling. This method was developed by the Caldecott lab to follow the maturation of Okazaki fragments[37,38]. In this assay, the EdU tail moment is driven by the presence of Okazaki fragments and other primed DNA structures generated during new DNA synthesis while the tail moment for total DNA reflects the induction of DNA damage. Low concentrations of POLAi (up to 0.5 μM) modestly inhibited the incorporation of EdU (Supplementary Fig. 2B) without changing the levels of priming (Supplementary Fig. 2C) and without inducing DNA damage (Supplementary Fig. 2D). At higher concentrations a strong block in DNA replication (Supplementary Fig. 2B) is accompanied with an increase in both EdU and total DNA tail moment (Supplementary Fig. 2C, D) that reflects the induction of DNA damage, in line with previous reports[36,39]. Thus, we decided to use a concentration of POLA1i that partially inhibits POLA1, leading to a moderate accumulation of ssDNA without eliciting DNA damage[36]. As a result of the partial inhibition of POLA1, the firing of new origins was reduced, leading to an increase in the speed of ongoing replication forks (Fig. 2E, F). Concomitant inhibition of POLA and VCP/p97 only led to a minor decrease in fork rate (Fig. 2E) and had no effect on DNA replication initiation (Fig. 2F), indicating that both proteins operate in the same pathway. The fact that VCPi did not prevent the increase in fork speed elicited by POLAi while it blocked the effect of CDC7i suggests that POLA/PRIM is also relevant for the effect of VCP/p97 during DNA replication elongation.

POLA inhibition has been reported to uncouple the progression of the leading and lagging strands[39]. In line with these results, fork and origin asymmetry were increased by POLAi to a similar extent than VCPi (Fig. 2G, H and Supplementary Fig. 2E–I). The simultaneous inhibition of POLA and VCP/p97 did not elicit any further increase in asymmetry measured through the ratio CldU/IdU in a single track, the ratio of long/short fork length from a single origin or the percentage of asymmetric origins (Fig. 2G-H and Supplementary Fig. 2E–I), again pointing to an epistatic relation of VCP/p97 and POLA in the control of DNA replication. In further support of this idea, both VCPi and POLAi suppressed the accelerated fork progression induced by CDC7i (Fig. 2C, I) without further reducing origin firing (Fig. 2D, J).

To further characterize the functional cooperation between VCP/p97 and POLA during DNA replication, we analyzed the activation of the RSR when both proteins were blocked in synchronized HCT116 cells released for 2 h in S phase. As reported[39], POLA inhibition led to a dose-dependent activation of ATR leading to CHK1 phosphorylation (Fig. 2K, lanes 1–5). As stated above, higher doses of POLAi generated DNA damage as measured by the increase in γH2AX and phosphorylated RPA2, consistent with the induction of replication catastrophe[39] (Fig. 2K, lanes 4–5). Inhibiting VCP/p97 in the presence of POLA inhibitors did not prevent the induction of damage and led to stronger phosphorylation of CHK1 even at low concentrations of POLAi (Fig. 2K, compare lanes 1–5 and 6–10; Supplementary Fig. 2J). None of the inhibitor treatments changed the levels of VCP/p97 or POLA1, and the inhibition of VCP/p97 only led to a slight accumulation of ubiquitylated proteins in these conditions (Supplementary Fig. 2K). These data indicate that VCPi does not prevent RPA exhaustion and replication catastrophe caused by POLAi. Instead, POLA/PRIM cooperates with VCP/p97 to regulate ATR activity and CHK1 phosphorylation during unperturbed DNA replication.

## VCP/p97 extracts POLA/PRIM from chromatin

Early work by the Cimprich lab proved that the accumulation of DNA primers generated by POLA/PRIM promotes TOPBP1 loading on chromatin and stimulates the activation of ATR[40]. Since the inhibition of VCP/p97 and POLA synergize in the activation of CHK1, we asked whether VCP/p97 could directly extract POLA/PRIM to limit the activation of the RSR. The potential interaction between POLA/PRIM and VCP/p97 was tested in immunoprecipitation assays in extracts from synchronized HCT116 cells released in S phase for 2 h and treated with VCPi or POLAi for 3 h. First, we carried out reciprocal pull-down experiments with antibodies against VCP/p97 and PRIM2 (PRIM2 was chosen because the IP of other components of the POLA/PRIM complex was either ineffective or disrupted the complex). PRIM2 IP co-precipitated VCP/p97 and the interaction was increased after VCPi treatment (Supplementary Fig. 3A), as expected, because the binding of VCP/p97 to its substrates is stabilized by substrate-trapping[41,42]. As a negative control, none of the pull-downs contained RPA2 (Supplementary Fig. 3A). However, VCP/p97 pull-down recovered no PRIM2 (Supplementary Fig. 3A). This result is not surprising, because VCP/p97 is a very abundant protein with a high number of substrates that are indirectly recognized through a number of cofactors.

In chromatin, VCP/p97 associates to the heterodimer UFD1L/NPLOC4 that serves as an adaptor to recognize ubiquitylated targets[43]. Thus, we surmised that the pull-down of these adaptors could recover the POLA/PRIM complex more efficiently. The immunoprecipitation of PRIM2 confirmed the interaction with VCP/p97 and also recovered UFD1L (Fig. 3A and Supplementary Fig. 3B). Compared to the input, the pull-down of PRIM2 recovers higher amounts of UFD1L than of VCP/p97 (Fig. 3A). Reciprocally, UFD1L pull-down recovered PRIM2 and POLA1 in both control and VCPi treated conditions (Fig. 3B and Supplementary Fig. 3C). The interaction between POLA/PRIM and UFD1L was not affected by the treatment with VCPi. As a negative control, SUMO-conjugating protein UBC9 was not recovered in any of the conditions (Fig. 3A, B). Together, these results show that VCP/p97 interacts with the POLA/PRIM complex and this interaction is mediated, at least in part, by the adaptor UFD1L.

Next, we tested if VCPi leads to the accumulation of POLA/PRIM on chromatin during S phase. As expected, the levels of several replication factors in chromatin slightly decreased when cells advance through S phase, including the processivity factor PCNA, components of the MCM helicase (MCM5), or part of the POLA/PRIM complex (Fig. 3C, DMSO-treated control cells). VCPi induced the accumulation of VCP/p97 itself on chromatin (Fig. 3C, VCPi-treated cells), in agreement with previous results[18]. In addition, VCPi led to the fast accumulation of POLA2 and PRIM2 on chromatin (Fig. 3C, VCPi). This was not a general effect on the replication machinery since the levels of PCNA, POLD1, MCM5 or CDC45 on chromatin were not affected (Fig. 3C). Interestingly, no consistent accumulation of POLA1 and only a slight increase in the levels of PRIM1 were observed after VCPi treatment (Fig. 3C). In contrast, immunofluorescence staining after pre-extraction of soluble proteins revealed that POLA1 accumulates on chromatin upon VCPi treatment (Fig. 3D, control in Supplementary Fig. 3D). These results suggest that ubiquitylated POLA1 accumulates on chromatin when VCP/p97 is inhibited but cannot be detected by Western blot, raising the possibility that VCP/p97 is targeted to the POLA/PRIM complex through the interaction with ubiquitylated POLA1. Thus, we analyzed if the interaction of UFD1L with POLA/PRIM is affected by the depletion of POLA1. The interaction of UFD1L and PRIM2 was reduced in cells depleted of POLA1 even if the levels of PRIM2 were not affected by the knockdown of POLA1 (Supplementary Fig. 3E). Last, we examined the ubiquitylation of POLA/PRIM through the purification of ubiquitylated proteins in HCT116 cells transfected with His-tagged ubiquitin[44]. Ubiquitylated POLA1 and PRIM2 were detected in HCT116 cells synchronized in G1/S and released for 2 h prior to the treatment with VCPi for 2 additional hours (Supplementary

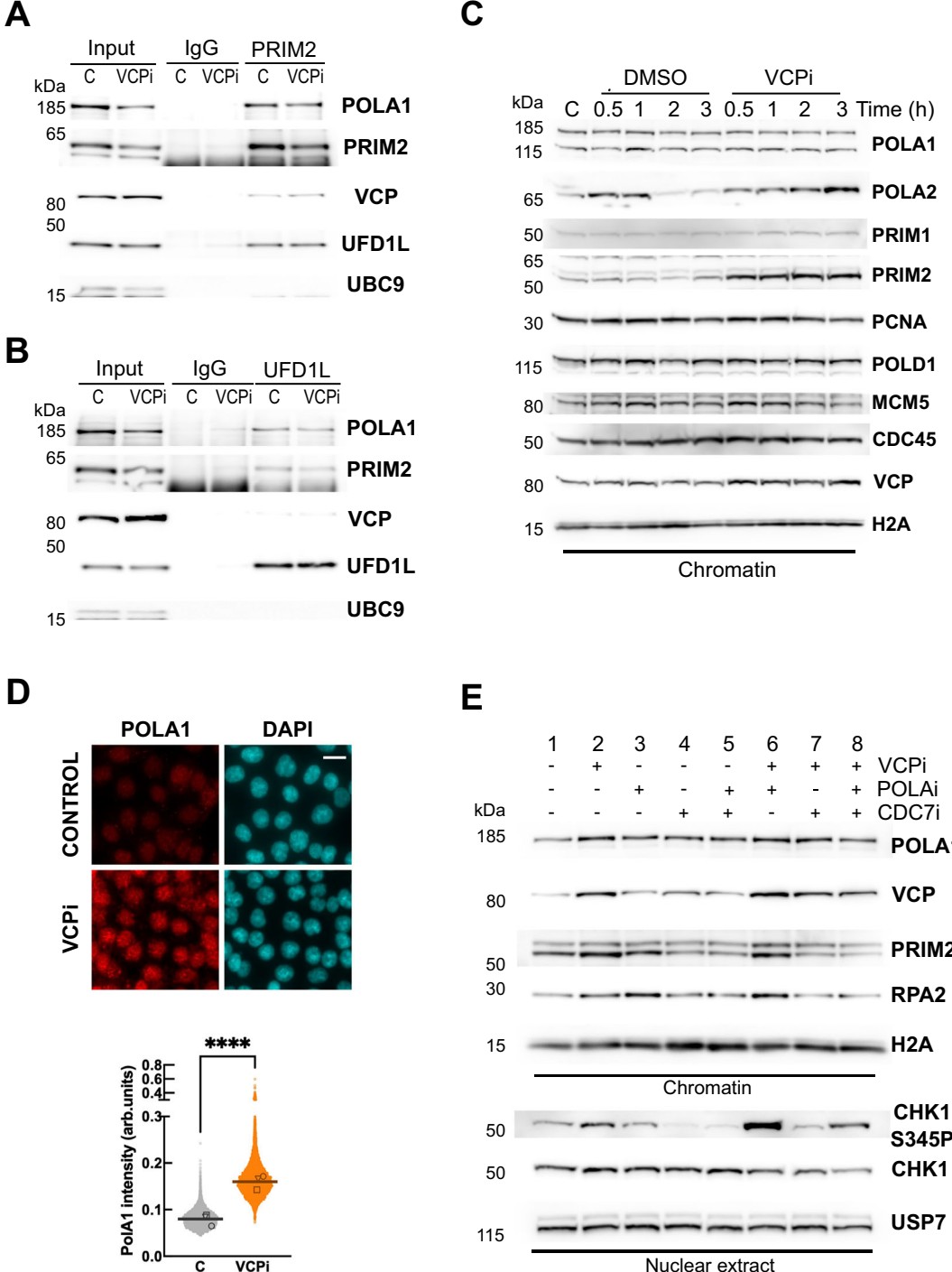

**Fig. 3 | VCP/p97 extracts POLA/PRIM to limit the replication stress response activation upon origin firing. A**, **B** Western blot analysis of the immunoprecipitation of PRIM2 (**A**) and UFD1L (**B**) from whole nuclear extracts from HCT116 cells, synchronized, released for 2 h and treated for 3 h with 5 µM NMS873 (VCPi) or DMSO as a control (marked as C). 2% of the input material is shown and a non-specific IgG was used as a negative control. The levels of POLA1, PRIM2, VCP/p97 and UFD1L were analyzed with specific antibodies; UBC9 was used as control. One representative experiment out of 4 replicates is shown. **C** Western blot analysis of the chromatin fraction from HCT116 cells, synchronized, released for 2 h (marked as C) and treated for the indicated times in the presence of 5 µM NMS873 (VCPi) or DMSO (DMSO). The levels of POLA1, POLA2, PRIM1, PRIM2, PCNA, POLD1, MCM5, CDC45 and VCP/p97 were analyzed with specific antibodies; histone H2A was used as control. One representative experiment out of 3 replicates is shown. **D** Immunofluorescence

analysis in HCT116 cells, synchronized, released for 2 h and treated for 2 h with 5 µM NMS873 (VCPi) or DMSO (Control). After pre-extraction of the soluble nuclear material, POLA1 was detected with specific antibodies. Nuclei were counterstained with DAPI and the scale bar indicates 25 µM. The graph shows the pool of three experiments, the median for the pooled data and the individual medians noted with different symbols. ****$p < 0.0001$ in Mann–Whitney test. **E** Western blot analysis of the chromatin and nuclear soluble fractions from HCT116 cells, synchronized, released for 2 h and treated for 2 h with 5 µM NMS873 (VCPi), 0.5 µM adarotene (POLAi), 20 µM XL413 (CDC7i), combinations of these drugs or DMSO as a control. The levels of POLA1, PRIM2 and total RPA2 were analyzed in chromatin using histone H2A as a control; the levels of total and phosphorylated CHK1 (S345) were analyzed in the nuclear soluble fraction using USP7 as a control. One representative experiment out of 7 replicates is shown. Source data are provided as a Source Data file.

Fig. 3F). Interestingly, both POLA1 and PRIM2 ubiquitylation was slightly increased by VCP/p97 inhibition (Supplementary Fig. 3F).

In vitro DNA replication experiments using *Xenopus* egg extracts have shown that POLA accumulation on chromatin and the subsequent activation of ATR requires origin firing but is also stimulated by the synthesis of new primers ahead of the stalled fork[40,45,46]. Thus, we investigated if the activation of CHK1 upon VCP/p97 inhibition was related to repriming ahead of stalled forks and/or to priming actions upon new origin firing. Repriming is mediated by the primase/polymerase PRIMPOL in the leading strand in the presence of DNA damage[47–49] while POLA/PRIM is responsible for the repriming in the lagging strand[50,51]. Genetic depletion of PRIMPOL[52] did not change the effect of VCPi on DNA replication (Supplementary Fig. 4A), arguing against an effect of VCP/p97 inhibition through PRIMPOL-mediated repriming. In contrast, inhibition of origin firing with CDC7i led to a slight rescue in EdU incorporation compared to the combined treatment of VCPi and POLAi (Supplementary Fig. 4B, lower row, compare first and last samples), suggesting that the effect of VCP/p97 on DNA replication is partially dependent on new origin firing. In line with this result, the inhibition of CDC7 reduced the accumulation of VCP/p97 and PRIM2 on chromatin caused by VCPi (Fig. 3E, lane 2 vs 7); this effect was stronger when CDC7 was inhibited in cells treated with a combination of VCPi and POLAi (Fig. 3E, lane 6 vs 8). As expected, the inhibition of CDC7 also reduced the accumulation of RPA2 on chromatin induced by POLAi (Fig. 3F, lanes 3 vs 5 and 6 vs 8). As a control, the levels of histone H2A and USP7 were not altered (Fig. 3E). Interestingly, the reduction in the amount of POLA/PRIM on chromatin was mirrored by a decrease in the phosphorylation of CHK1 in the soluble nuclear fraction (Fig. 3E, lane 2 vs 7, and lane 6 vs 8; see quantification in Supplementary Fig. 4C). A recent report shows that CDC7 activity is required for CHK1 activation during the unperturbed S phase[53], linking the activation of the RSR to the number of active replication forks and, potentially, to the accumulation of POLA/PRIM on chromatin. In our system, CDC7i only partially rescued the increase in CHK1 phosphorylation induced by the combination of VCPi and POLAi. Thus, the effect of VCPi on the RSR is not exclusively due to newly fired origins and is also related to the control of ongoing DNA replication forks.

ATR activation is a two-step process mediated by the recruitment of the ATR-ATRIP complex to ssDNA by RPA and its activation through specific domains in TOPBP1 or ETAA1. While TOPBP1 is recruited to ssDNA/dsDNA junctions and requires the presence of primed DNA, ETAA1 directly binds to the RPA complex and its action is linked to the accumulation of ssDNA. Interestingly, the phosphorylation of CHK1 caused by VCPi in synchronized HCT116 cells was dependent on the presence of TOPBP1 (Supplementary Fig. 4D, siTOPBP1), while the depletion of ETAA1 using a previously validated siRNA[39] did not have any effect on the activation of ATR by VCP/p97 inhibition (Supplementary Fig. 4D, siETAA1). We conclude that VCP/p97 controls the recruitment of POLA/PRIM to chromatin that could in turn determine the generation of primed DNA that serves as a substrate for TOPBP1-mediated ATR activation.

**POLA/PRIM depletion confirms the involvement of VCP/p97 in DNA priming**

We combined validated siRNAs directed against the 4 subunits of POLA/PRIM[39] to induce the depletion of the whole complex while minimizing off-target effects by using low concentrations of each siRNA. We observed a very efficient reduction in the levels of POLA1, PRIM1 and PRIM2, and a mild reduction in POLA2 (Supplementary Fig. 5A). Cells depleted of POLA/PRIM were synchronized with a single thymidine block and released for 2 h into S phase before the treatment with VCPi for 2 h. The depletion of POLA/PRIM increased fork speed (Fig. 4A), strongly impaired origin firing (Fig. 4B) and greatly increased CHK1 and RPA2 phosphorylation (Fig. 4C), indicating that ssDNA is accumulated in POLA/PRIM-depleted cells leading to a strong

activation of ATR/CHK1. Blocking the action of VCP/p97 did not further affect new origin firing in cells with reduced levels of POLA/PRIM (Fig. 4B) but it reduced fork speed in these cells (Fig. 4A). These results are in line with the observation that the POLA/PRIM complex acts as a brake for DNA replication in model viral systems[54]. Further, the strong activation of CHK1 induced by POLA/PRIM downregulation could also contribute to promote fork progression through the activation of pathways such as translesion synthesis[31,55]. In contrast, the partial inhibition of POLA1 in a short period of time would induce fork acceleration through alternative mechanisms that do not rely on the activity of VCP/p97 (Fig. 2E).

Interestingly, the inhibition of VCP/p97 did not increase CHK1 phosphorylation in POLA/PRIM- depleted cells as it did in control transfected cells (Fig. 4C, D), and in contrast to the cumulative phosphorylation of CHK1 induced by the combination of VCPi and POLAi (Fig. 2K). In POLA/PRIM-depleted cells, VCPi would still increase priming by the retention of POLA/PRIM and lead to higher CHK1 phosphorylation. At the same time, VCPi decreases the fork rate and the levels of ssDNA, as reflected by the reduced phosphorylation of RPA2. As a result, no net changes in CHK1 phosphorylation were observed. Finally, the depletion of the POLA/PRIM complex led to increased fork asymmetry (Fig. 4E), long/short fork length ratios (Fig. 4F and Supplementary Fig. 5B–E) and to an elevated percentage of asymmetric origins (Supplementary Fig. 5F). Again, VCPi increased fork and origin asymmetry in control cells, but it did not further increase the effects of the depletion of POLA/PRIM in fork asymmetry (Fig. 4E), the long/short fork length ratio (Fig. 4F and Supplementary Fig. 5B–E) or the percentage of asymmetric origins (Supplementary Fig. 5F). These experiments confirmed that VCP/p97 works together with POLA/PRIM in the control of origin firing and fork symmetry.

**VCP/p97 limits CHK1 activation through the control of POLA/PRIM**

Our results support a model where VCP/p97 limits the amount of POLA/PRIM on chromatin during DNA replication and restricts its priming activity. According to this model, VCP/p97 would restrict the activation of ATR and CHK1 by limiting the number of primed DNA structures (Fig. 5A). In an unperturbed S phase VCP/p97 would buffer the basal level of ATR activation (Fig. 5A, top) that is required to prevent the premature firing of late S phase origins. In this model, VCPi would lead to the accumulation of POLA/PRIM on chromatin, the subsequent activation of ATR by TOPBP1 and the phosphorylation of CHK1 (Fig. 5A, bottom). Thus, the effects of VCP/p97 on DNA replication dynamics should be direct and not a consequence of the activation of CHK1. The depletion of CHK1 has been shown to reduce fork speed and stimulate new origin firing through independent mechanisms[31]. Similarly, the inhibition of CHK1 strongly reduced fork speed and induced new origin firing (Supplementary Fig. 6A, B). While the concomitant inhibition of CHK1 and VCP/p97 did not reduce fork speed further (Supplementary Fig. 6A), VCPi decreased origin firing even if CHK1 was also inhibited (Supplementary Fig. 6B). We conclude that the decrease in origin firing elicited by VCPi is not due to the activation of CHK1 but it represents a direct role for VCP/p97 in the control of origin firing.

If the generation of primed DNA by POLA/PRIM underlies the activation of CHK1 by VCPi, then the complete inhibition of POLA1 should prevent this activation. To address this point, we treated synchronized HCT116 cells released into S phase for 2.5 h with increasing concentrations of POLA1i in combination with the inhibition of VCP/p97. Interestingly, higher concentrations of POLAi led to a lower phosphorylation of CHK1, even if there was a further increase in RPA phosphorylation (Fig. 5B, compare lanes 3 and 5). Similarly, while CHK1 phosphorylation was strongly induced by the combination of VCPi with low levels of POLAi, the complete block of POLA1 activity returned the levels of CHK1 phosphorylation to those

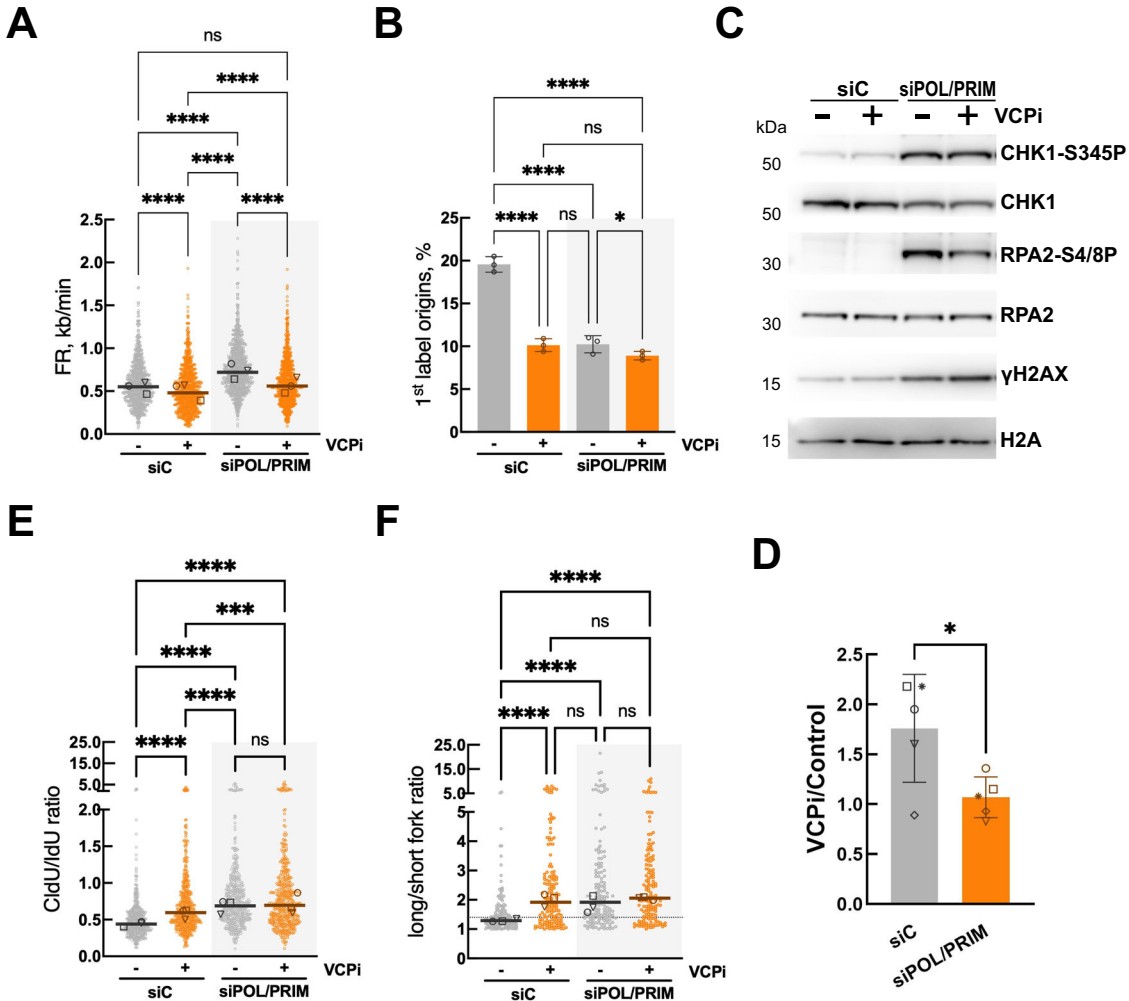

**Fig. 4 | VCP/p97 cooperates with POLA/PRIM to control DNA replication and the replication stress response.** HCT116 cells were transfected with a non-specific siRNA (siC) or a combination of individual siRNA directed against POLA1, POLA2, PRIM1 and PRIM2. 24 h after transfection cells were synchronized, released for 2 h and treated with 5 µM NMS873 (VCPi) for 2 h or DMSO as control. **A**, **B** DNA fiber analysis of HCT116 cells depleted of POLA/PRIM. FR (**A**) and 1st label origins (**B**) were determined. **C** Western blot analysis of whole cell extracts measuring the levels of total and phosphorylated CHK1 (S345), total and phosphorylated RPA2 (S4/S8), phosphorylated H2AX (γH2AX), VCP/p97 and histone H2A as control. The experiments were repeated five times and one representative result is shown. **D** Densitometric quantification of the levels of phosphorylated CHK1 (S345)

normalized to the levels of total CHK1 in 5 independent experiments from Fig. 4C. Mean ± SD is plotted; *, $p = 00278$ in t-test. **E**, **F** DNA fiber analysis in HCT116 cells depleted of POLA/PRIM. CldU/IdU ratio (**E**) and long/short ratio (**F**) were determined. Individual data for the long/short ratio are shown in Supplementary Fig. 5B-E. For all DNA fiber analysis three independent experiments were performed; dot plots show pooled data and bars represent the median (FR, CldU/IdU and long/ short fork) and bar graphs (1st label origins) plot the mean ± SD. The median of individual experiments is noted with different symbols both in dot plots and bar graphs. *$p < 0.05$; ****$p < 0.0001$; ns, non-significant, in Kruskall–Wallis with Dunn's post-test (**A**, **E**, **F**) or one-way ANOVA followed by Tukey's test (**B**). Source data are provided as a Source Data file.

observed in the absence of VCPi (Fig. 5B, compare lanes 6 and 8) independently of the phosphorylation of RPA2. These results are consistent with the complete inhibition of POLA1 inducing replication catastrophe due to a toxic accumulation of ssDNA[39,56] while limiting the activation of ATR by preventing the formation of primed DNA structures. We conclude that the activation of CHK1 upon VCP/ p97 inhibition requires the generation of primers by POLA/PRIM, most likely in the lagging strand (Fig. 5A).

One of the main functions of CHK1 phosphorylation is the activation of the G2/M checkpoint to prevent the entry into mitosis in the presence of unreplicated or damaged DNA. Previous results have shown that VCP/p97 inhibition leads to a cell cycle arrest in G2/M[30] that has been linked to the inhibition of replisome disassembly after DNA replication termination[57]. Our results raise the possibility that the increase in CHK1 phosphorylation induced by VCPi during S phase directly activates the G2/M checkpoint. In agreement with these data, VCPi treatment led to the cell cycle arrest in G2/M (Fig. 5C) that is

partially bypassed by the inhibition of CHK1, allowing the cells to progress through mitosis into G1 even in the presence of VCPi (Fig. 5C).

## VCP/p97 regulates the priming activity of POLA/PRIM

The key point in our model is that VCP/p97 controls priming during DNA replication. Using the EdU comet assay[37,38] we confirmed our observations (Supplementary Fig. 2B–D) that partial inhibition of POLA1 did not change the EdU tail moment (Fig. 6A, B) indicating that, at this concentration, POLAi only minimally affects priming in the lagging strand. On the other hand, the depletion of POLA/PRIM reduced the EdU tail moment (Fig. 6C, D), confirming that limiting levels of POLA/PRIM reduced the frequency of priming in the lagging strand as proposed by Porcella et al.[58]. Neither the low-concentration POLAi or the depletion of POLA/PRIM changed the total DNA tail moment (Supplementary Fig. 7A, B). In contrast, the complete inhibition of POLA/PRIM with 5 µM POLAi increased both EdU and total DNA tail moments (Supplementary Fig. 7C, D) as we showed in

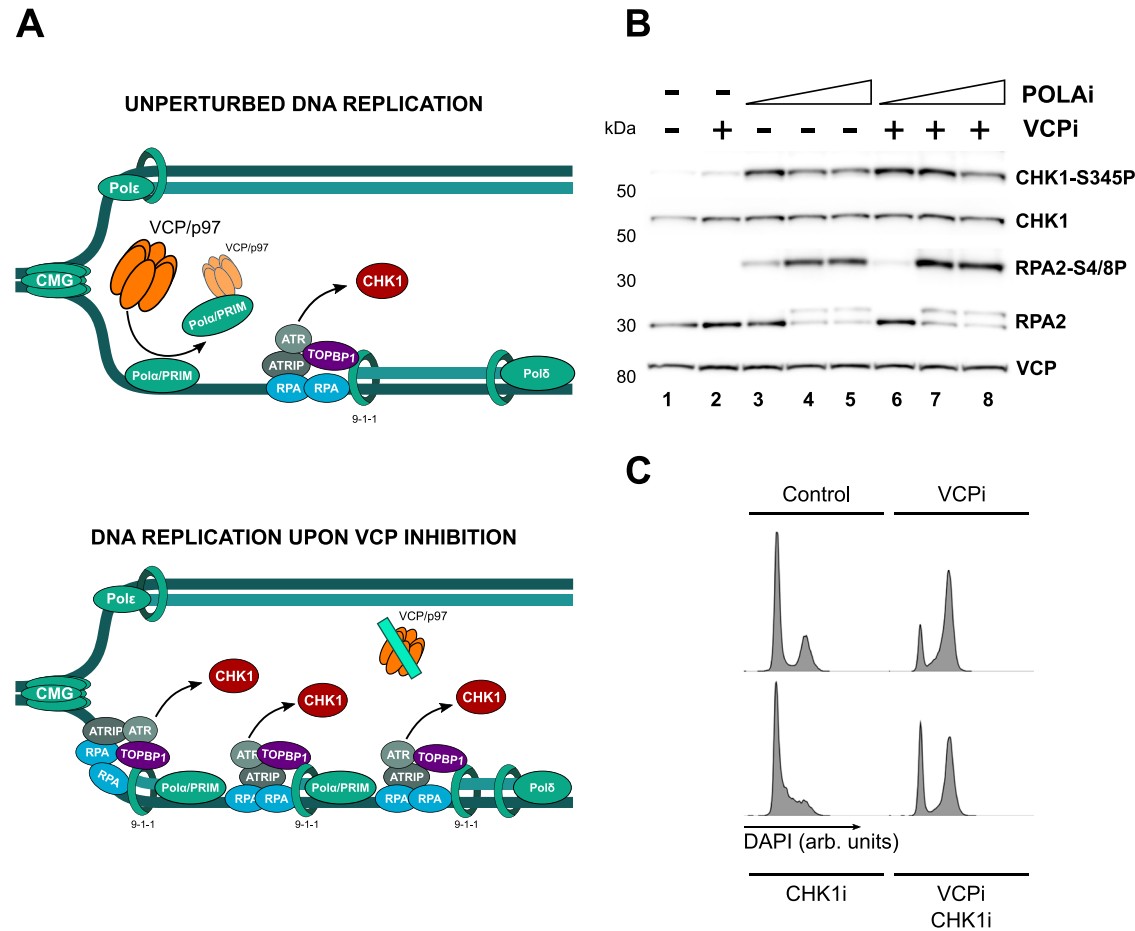

**Fig. 5 | VCP/p97 activates the RSR through POLA/PRIM. A** Model for the action of VCP/p97 during an unperturbed S phase. POLA/PRIM is loaded to chromatin and generates primed DNA that activates ATR in a TOPBP1 dependent manner. VCP/p97 extracts POLA/PRIM from the lagging strand to limit the activation of CHK1.
**B** Western blot analysis of whole extracts purified from HCT116 cells, synchronized, released for 2.5 h, treated for 10 min with increasing concentrations of adarotene (POLAi), and then treated for 2 h with 5 µM NMS873 (VCPi), increasing concentrations of adarotene (POLAi), a combination of these drugs or DMSO as a control

(marked as double negative). The levels of total and phosphorylated Chk1 (S345), total and phosphorylated RPA2 (S4/S8), and VCP/p97 were measured with specific antibodies. The experiment was repeated three times and one representative result is shown. **C** Flow cytometry analysis of the cell cycle progression showing histograms for DNA content by DAPI staining in HCT116 cells, synchronized, released for 3 h and treated for 9 h with 5 µM NMS873 (VCPi), 2.5 µM LY2603618 (CHK1i), a combination of both (VCPi-CHK1i) or DMSO as a control. Source data are provided as a Source Data file.

Supplementary Fig. 2B–D. The increase in total DNA tail moment supports the induction of replication catastrophe leading to widespread DNA breaks. Under these conditions the increase in the EdU tail moment likely mirrors the increase in the tail moment of total DNA (Supplementary Fig. 7C, D). Together, these data validate the use of the EdU comet assay to measure the priming activity of POLA/PRIM in unperturbed DNA replication.

Next, we evaluated the effect of the inhibition of VCP/p97 on the generation of primed DNA by POLA/PRIM. In agreement with our model, the EdU tail moment was increased by VCPi treatment (Fig. 6E, F). In addition, VCPi also increased the EdU tail moment when combined with partial POLA1 inhibition (Fig. 6G, H) and in POLA/PRIM depleted cells (Fig. 6I, J). Importantly, the tail moment for total DNA was not significantly changed by the treatment with VCPi alone or in combination with POLAi (Supplementary Fig. 7E, F). A minor increase in the tail moment for total DNA is observed after the treatment of POLA/PRIM depleted cells with VCPi (Supplementary Fig. 7G). We conclude that the inhibition of VCP/p97 increases priming by POLA/PRIM. Combined, these data suggest that VCP/p97 limits priming during an unperturbed S phase by removing POLA/PRIM complexes from chromatin, keeping the basal activity of ATR in check and preventing the activation of the G2/M checkpoint.

## Discussion

The mechanisms of activation of the RSR have been mainly elucidated employing exogenous sources of replication stress that promote the uncoupling of the CMG helicase and the replicative DNA polymerases, leading to the accumulation of ssDNA[2]. However, ATR is also activated during an unperturbed S phase and is essential to sustain the full and accurate replication of the cellular DNA in the absence of damage[5]. Interestingly, the subset of targets phosphorylated by ATR during S phase differs from the proteins modified by ATR in the presence of replication stress[59,60].

Initial reports showing that ATR and CHK1 inhibition accelerated DNA replication led to the suggestion that the basal activation of the RSR restricts origin firing to prevent the depletion of resources necessary during DNA replication[61,62]. Later it was shown that the activation of a limited number of ATR molecules prevents the phosphorylation of MCM4 and the GINS complex by CDC7 to inhibit new origin firing both locally and in late S phase[63,64]. Inhibiting ATR in the absence of damage increases the number of active origins and replication forks, at the price of decreased fork speed and an accumulation of collapsed forks[65]. In line with these results, recent work has shown that the deletion of ATR does not affect DNA replication initiation but it leads to replication failure by mid S phase in B cells. In this model,

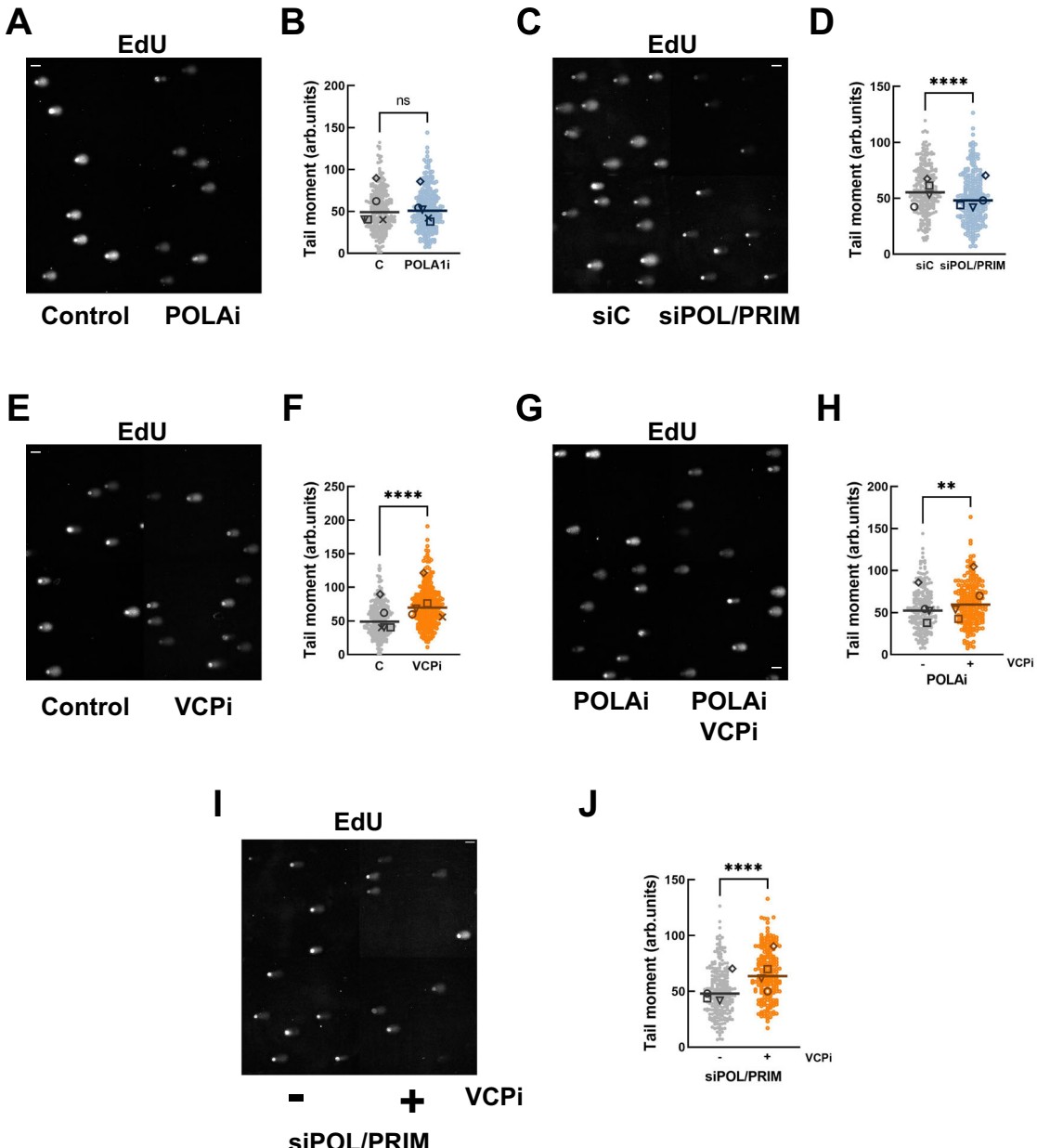

**Fig. 6 | VCP/p97 limits priming by POLA/PRIM.** Comet assay on nascent DNA labelled with EdU. Cells were incubated with 30 μM EdU for 60 min, and EdU was conjugated to a fluorescent probe after alkaline comet assay. **A, B** HCT116 cells were synchronized, released for 2.5 h and treated for 2 h with 0.5 μM adarotene (POLAi) or DMSO as a control. Representative images are shown in (**A**) and quantification in (**B**). **C, D** HCT116 cells were transfected with a non-specific siRNA (siC) or a combination of individual siRNA directed against POLA1, POLA2, PRIM1 and PRIM2. 24 h after transfection cells were synchronized, released for 3 h and incubated with EdU. Representative images are shown in (**C**) and quantification in (**D**). **E, F** HCT116 cells were synchronized, released for 2.5 h and treated for 2 h with 5 μM NMS873 (VCPi) or DMSO as a control. Representative images are shown in (**E**) and quantification in (**F**). **G, H** HCT116 cells were synchronized, released for 2.5 h and treated for 2 h with 0.5 μM adarotene (POLAi) alone or in combination with 5 μM NMS873 (VCPi). Representative images are shown in (**G**) and quantification in (**H**). **I, J** HCT116 cells were transfected with a combination of individual siRNA directed against POLA1, POLA2, PRIM1 and PRIM2. 24 h after transfection cells were synchronized, released for 2.5 h HCT116 cells and treated for 2 h with 5 μM NMS873 (VCPi) or DMSO as a control. Representative images are shown in (**I**) and quantification in (**J**). In all cases the dot plots show pooled data of at least 4 experiments and bars represent the median of pooled data. The individual medians of independent experiments are noted with different symbols. **$p < 0.01$; ****$p < 0.0001$; ns, non-significant, in Mann–Whitney test (**B**, **D**, **F**, **H**, **J**). Source data are provided as a Source Data file.

blocking origin firing rescues the nucleotide depletion and prevents the DNA damage induced by the lack of ATR[8]. Thus, the basal ATR activity limits the number of origins that are activated simultaneously in the absence of replication stress, to ensure the correct progression of DNA replication and prevent the depletion of replication factors and nucleotides. Although a recent work has linked the basal level of ATR activity to the accumulation of RPA during DNA replication[7], the mechanisms that set the basal ATR activity remain largely unknown.

Here we provide evidence that the basal activation of ATR is driven by the loading of POLA/PRIM on chromatin and the generation of primed DNA, and is restricted by the extraction of this complex by VCP/p97. This model is supported by different lines of evidence. First, VCPi activates ATR in a TOPBP1-dependent manner (Supplementary Fig. 4). Since TOPBP1 is loaded by the presence of primed DNA structures, this suggests that primed DNA structures are accumulated when VCP/p97 is blocked. Second, the activation of CHK1 by VCPi requires

POLA/PRIM catalytic activity, since the complete inhibition of POLA1 abrogates the activation of CHK1 in the presence of VCPi (Fig. 5)[36]. Third, VCPi leads to the accumulation of Okazaki fragments and primed DNA, proving that the extraction of POLA/PRIM by VCP/p97 restricts priming, most likely in the lagging strand (Fig. 6). Last, the inhibition of origin firing partially prevents CHK1 activation induced by VCPi (Fig. 3F and Supplementary Fig. 4C), showing that VCP/p97 controls priming after origin firing and in ongoing DNA replication forks. We note that, in this context, priming events were linked to POLA/PRIM and not to PRIMPOL, as genetic ablation of the latter did not alter the effect of VCP/p97 inhibition on DNA replication (Supplementary Fig. 4A), in line with a recent report showing that the activation of re-priming by PRIMPOL requires its phosphorylation by CHK1 that is only achieved through the overexpression of Claspin that further activates CHK1[66].

Recent structural and biochemical data have shed light on the mechanisms of priming by the POLA/PRIM complex during DNA replication. In addition to the previously reported interaction with Ctf4/AND1[67], the POLA/PRIM complex directly associates with the CMG helicase through the interaction with MCM proteins and is positioned to accommodate the lagging strand directly after unwinding[68]. This conformation allows the synthesis of the RNA primer that is elongated by POLA1 as the replisome advances[69,70]. The stability of the association between POLA/PRIM and CMG is not clear. In fact, limiting the amount of POLA/PRIM leads to longer Okazaki fragments, arguing in favor of a distributive model where priming by POLA/PRIM involves the association/dissociation of the complex from chromatin[58]. In agreement with the distributive model, we show that the regulation of POLA/PRIM association with chromatin limits DNA priming during replication. Our data also support the idea that the number of primed DNA structures is an important factor to set the basal level of activation of ATR/CHK1. In fact, the inhibition of Okazaki fragment maturation has been previously shown to activate the RSR by inducing the accumulation of primed DNA[71]. We show that VCP/p97 extracts POLA/PRIM from chromatin in unperturbed conditions, restraining the generation of primed DNA structures and preventing an excessive activation of ATR that would impair the progression through the cell cycle (Fig. 5A). We cannot discard an additional contribution of VCP/p97 to the maturation of Okazaki fragments and this would be a very interesting question for future experiments.

Apart from the synthesis of primers in the lagging strand during DNA replication, POLA/PRIM also takes part in the extension of telomeres as part of the CST complex[72-74], in the fill-in activity during NHEJ[75], in the inheritance of histones during DNA replication in cooperation with MCM2 and AND1[76-78], and is also recruited to stalled forks through the interaction with RAD51[50]. Our results on the accumulation of POLA/PRIM on chromatin upon VCP/p97 inhibition suggest that only a fraction of the complex is controlled by VCP/p97-mediated extraction, supporting the existence of different mechanisms of association of POLA/PRIM to chromatin. Similarly, Ctf4/AND1 only affects Okazaki fragment length when the amounts of POLA/PRIM are reduced and is dispensable for the interaction of the complex with the CMG helicase, indicating that there are specific mechanisms that recruit POLA/PRIM to chromatin in certain contexts[58]. We hypothesize that the stable association to the replisome could be required for the correct inheritance of chromatin states, while a more mobile fraction of POLA/PRIM that is controlled by VCP/p97 enables the dynamic modulation of the number of Okazaki fragments generated during DNA replication.

Based on our model the inhibition of VCP/p97 should induce an accumulation of primed structures and the subsequent phosphorylation of CHK1 (Fig. 5B), activation of the checkpoint and the arrest of the cell cycle in G2/M. It has been previously proposed that the G2/M arrest induced by VCPi is related to the accumulation of replication proteins on chromatin[57] due to the blockage of replisome disassembly

after DNA replication termination[79]. In this sense, a recent study has shown that replisome disassembly in S phase is necessary to recycle DNA replication factors. Deleting the E3 ubiquitin ligase that marks the replisome for the extraction by VCP/p97 activates the G2/M checkpoint[57]. However, this study did not analyze if VCP/p97 inhibition itself would induce such an effect. We show that a short inhibition of VCP/p97 still activates the checkpoint without inducing a general accumulation of replication factors. Thus, checkpoint activation by a short treatment with VCPi is more likely the consequence of enhanced basal CHK1 phosphorylation due to increased priming, while the accumulation of unloaded replisomes in late S phase would be relevant after longer treatments.

Interestingly, our data also suggests that VCP/p97 directly controls fork progression. We show that VCP/p97 activity is necessary to sustain a physiological fork rate in a POLA/PRIM-dependent manner, and this effect could be mediated by several mechanisms. First, the removal of POLA/PRIM by VCP/p97 could facilitate the switch to DNA polymerases δ and ε to allow the progression from initiation to elongation. Second, VCP/p97 could be required to recycle other limiting replication factors, as suggested by recent work[57]. Last, VCPi could lead to the uncoupling of the leading and lagging strands. Further experiments will be required to fully understand how VCP/p97 determines DNA replication dynamics and to identify what cofactors participate in each of these processes.

The basal activation of the ATR/CHK1 pathway during S phase controls late S phase origins. In this sense, we had previously hypothesized that origin activation is the signal that locally prevents the firing of back-up origins[80]. Here, we provide evidence showing that, not only the number of active origins is important, but priming by POLA/PRIM also controls the level of basal ATR/CHK1 activity.

The effects of VCP/p97 inhibitors on DNA replication are important for their application in cancer treatment. The inhibition of VCP/p97 was initially proposed to induce cancer cell death through an increase in proteotoxic stress[81,82]. However, recent data indicate that the cytotoxic action of VCPi is related to the control of DNA replication and repair[27], by inducing an increase in PARP1 trapping[83] or altering DNA repair pathway choice[84-86]. Further, the anti-alcohol abuse drug disulfiram targets the VCP/p97 adaptor NPL4 and interferes with the RSR to prevent cancer development[87,88].

In summary, we propose that VCP/p97 limits the amount of POLA/PRIM on chromatin, restricting the generation of Okazaki fragments to set the basal activity of ATR/CHK1. Our results support a model where a pool of POLA/PRIM associates in a distributive manner with chromatin, serving as a sensor of the progression of DNA replication to modulate, in the absence of damage, the rate of origin firing and the speed at which S phase can progress in the absence of genomic instability.

## Methods

### Cell lines, extract preparation, transfections and treatments

HCT116 wild-type (ATCC, CCL-247) and PRIMPOL knockout (generated in the Méndez lab), and U2OS cells were grown in DMEM with 10% FBS, penicillin (100 IU/ml), streptomycin (100 mg/ml) and glutamine (300 mg/ml). For passaging, cells were washed once with warm PBS and trypsinized with Trypsin-EDTA. Trypsin was inactivated by the addition of fresh media and the cell suspension was centrifuged at 800 g for 5 min. All cell culture reagents were prepared by the cell culture facility in CBM.

NMS873 (Tocris, 6180), adarotene/ST1926 (MedChemExpress, HY-14808), XL413 (Selleckchem, BMS-863233), LY2603618 (Selleckchem, S2626) were dissolved in dimethyl sulfoxide (DMSO); hydroxyurea (Sigma-Aldrich, H8627) was dissolved in water. The treatment of cells was performed for the indicated time at the indicated concentration of inhibitors or an equivalent amount of DMSO. Cell synchronization was carried out by incubating the cells in the presence of 2 mM thymidine (Sigma-Aldrich, T1895) for 16 h. After that,

cells were washed twice with warm PBS and released for the indicated time in drug-free DMEM before the indicated treatments were added.

Whole cell extracts were prepared by adding 50 mM Tris, pH 7.5, 8 M Urea, and 1% Chaps to the cell pellet and vortexing for 30 min at 4 °C, followed by 5 min centrifugation at 20000 g and 4 °C. The supernatant was collected as whole cell extract. Subcellular fractionation was carried out resuspending cells in ice-cold 10 mM HEPES, pH 7.9, 10 mM KCl, 0.1 mM EDTA, incubating on ice for 10 min and then adding Nonidet P-40 to a final concentration of 0.1%. Nuclei were isolated by centrifugation at 2500 g for 5 min at 4 °C. The pellet was washed in 10 mM HEPES, pH 7.9, 10 mM KCl, 0.1 mM EDTA. The soluble nuclear fraction was extracted by vigorous shaking in 20 mM HEPES, pH 7.9, 0.4 M NaCl, 1 mM EDTA for 1 h at 4 °C followed by centrifugation at 16000 g for 5 min at 4 °C[89]. After the isolation of the cytosolic and nuclear extracts, the chromatin fraction was obtained adding 50 mM Tris, pH 7.5, 8 M Urea, and 1% Chaps to the pellet remaining after nuclear extraction.

Transfection of HCT116 cells with specific siRNA was carried out using Lipofectamine RNAimax (Invitrogen, 13778075) according to the manufacturer's instructions and using pools of 4 specific siRNA directed against POLA1, POLA2, PRIM1 or PRIM2 (Dharmacon, ON-TARGETplus SMARTPool; final concentration of 12.5 μM for each pool). As a control cells were transfected with All-stars control (Qiagen, 1027280). U2OS cells were transfected using DharmaFECT™ following manufacturer's guidelines with a mixture of two siRNA against ETAA1 (UGACAAAGCAGUUAGGUAA[dT][dT] and GAGCAAAACAAGAGGAAU U[dT][dT]) or a single siRNA against TOPBP1 (GGAUAUAUCUUUG CGGUUU[dT][dT]) using the following sequence as negative control (UAACGACGCGACGACGUAAtt). pCI-His-hUbi was a gift from Astar Winoto (Addgene plasmid # 31815; http://n2t.net/addgene:31815; RRID:Addgene_31815)

## Antibodies

The antibodies against USP7 (Bethyl A300-033A, 1/1000), VCP (Bethyl A300-589A and Santa Cruz Biotechnologies sc-57492, 1/1000), SUMO2/3 (MBL, M114-3 and University of Iowa, clone 8A2, 1/1000), PCNA (Santa Cruz, sc-56, 1/1000), Chk1 (Novocastra and Cell Signalling #2360, 1/500), Chk1-S345P (Cell Signaling #2348, 1/500), RPA2 (Abcam ab2175, 1/1000), RPA2-S4/S8P (Bethyl, A300-245A, 1/1000), H2A (Cell Signaling #3636, 1/1000), γH2AX (Millipore, 05-636, 1/1000), Ubiquitin (Cell Signaling #3933, 1/1000), MCM2 (custom made, Juan Méndez, 1/1000)[90], MCM2-S40P (Abcam, ab133243, 1/500), MCM2-S53P (Abcam, ab70367, 1/500), POLA1 (Abcam, ab31777, 1/1000 in WB, 1/250 in IF), PRIM2 (Invitrogen PA5-88189, 1/1000), TOPBP1 (Bethyl, A300-111, 1/1000), Vinculin (Sigma-Aldrich V9264, 1/1000), UFD1L (Abcam, ab155003), CDC25A (Santa Cruz Biotechnology sc-7389) were used for Western Blot and immunofluorescence. POLA1 (Abcam, ab31777, 1/250), CldU (Sigma-Aldrich, C6891, 1/100), IdU (Sigma-Aldrich, I7125, 1/100), anti-ssDNA antibody (Millipore MAB3034, 1/200), were used for immunofluorescence. VCP (Abcam, ab11433 and Santa Cruz Biotechnologies sc-57492, 3 mg of antibody for 100 mg of protein extract), PRIM2 (Invitrogen, PA5-88189, 3 mg of antibody for 100 mg of protein extract) and UFD1L (Abcam, ab155003, 1.5 mg of antibody for 100 mg of protein extract) antibodies were used for immunoprecipitation.

## Flow cytometry

For the analysis of the cell cycle, cells were incubated with 20 μM EdU (Sigma-Aldrich, 900584) for 30 min. Then, cells were trypsin-digested, washed with cold PBS once and fixed in 4% PFA/PBS for 15′ at room temperature (PFA, Electron Microscopy Services, 15710). After permeabilization with 0.25 % Triton/PBS for 20′ at room temperature, the EdU was labelled by a Click reaction with a fluorescent azide (AF488-Azide, Jena Bioscience, CLK-1275-1) in the presence of 10 mM Sodium Ascorbate, 2 mM CuSO4 in PBS for 30′ at room temperature. Then, the

DNA was stained with DAPI 0.5 μg/ml in the presence of 0.25 mg/ml RNAse A. All samples were analyzed in a BD LSRFortessa or in a FACSCanto II cell analyzer. The results were analyzed using the FlowJo software (FlowJo, LLC v10).

## DNA fiber assays

For the analysis of fork rate and origin firing, HCT116 cells were sequentially pulse-labeled with 50 μM CldU (Sigma-Aldrich, C6891) and 250 μM IdU (Sigma-Aldrich, I7125) for 20 min. For the analysis of fork symmetry, HCT116 cells were sequentially pulse-labeled with CldU for 10 min and IdU for 30 min. DNA fibers were spread in buffer containing 0.5% SDS, 200 mM Tris pH 7.4 and 50 mM EDTA. The CldU and IdU tracks were detected by IF with anti-BrdU antibodies (rat for CldU, Abcam ab6326, and mouse for IdU, BD 347580). Fiber integrity was assessed with a mouse anti-ssDNA antibody (Millipore MAB3034). Slides were examined with a DM6000 B Leica microscope with an HCX PL APO 40×, 0.75 NA objective[47]. The conversion factor used was 1 μm=2.59 kb[91]. Signals were measured and quantified using ImageJ software[92]. For FR, 250-350 forks (red-green tracks) were measured per condition in each independent experiment. For origin firing, origins labeled during the first pulse (green-red-green structures) were quantified as percentage of all structures containing red ( > 500 total structures scored in each case). For CldU/IdU ratio, red and green labels in 150 ongoing forks were measured. For terminations "only red" signals were quantified as percentage of all structures containing red ( > 500 total structures scored in each case)[93]. For origin asymmetry, both green tracks coming from a single origin were measured in 50 origins per condition in each independent experiment. An origin is considered asymmetric when the ratio between the two forks (long/short) is greater than 1.4.

## Immunoprecipitation

0.5-1 mg of protein were diluted at 1 mg/ml in BC200 (50 mM Tris pH 7.9, 200 mM NaCl), centrifuged for 5 min at 20,000 g at 4 °C and the supernatant collected as input material. After washing protein G Dynabeads (Invitrogen, 10004D) twice in BC200, the beads were incubated with anti-VCP antibody, anti-PRIM2 antibody or a non-specific IgG in the presence of 0.1 mg/ml BSA in BC200 for 30′ at 4 °C. Next, the Dynabeads were washed 5 times in BC200, incubated with the cleared supernatant ON at 4 °C and then washed five times with BC200 with 0.05% IGEPAL CA630 (Sigma-Aldrich, I8896). Bound material was eluted in loading buffer by heating for 10′ at 70 °C.

## Immunofluorescence analyses

Cells were cultured in polylysine-treated coverslips. To visualize chromatin-bound proteins, soluble factors were pre-extracted with 0.5% Triton X-100 in CSK buffer (10 mM PIPES pH 7.0, 0.1 M NaCl, 0.3 M sucrose, 3 mM MgCl2, 0.5 mM PMSF) for 10 min at 4 °C prior to fixation in 4% PFA (15 min/RT). Coverslips were incubated in blocking solution (3% BSA in PBS + 0.05% Tween 20) for 30 min. Primary (1:100 dilution) and secondary antibody (1:200 dilution) incubations were performed for 1 h/RT. Nuclei were stained with 0.1 μg/ml DAPI (Sigma-Aldrich, D9542) for 5 min. ProLong Gold antifade mounting media (Invitrogen, P36930) was used. Images were acquired in a Thunder Imaging System (Leica-Microsystems) equipped with AFC, LED8 excitation light source, HC PL APO 40x/NA 0.95 dry objective, and a DFC9000GTC camera, under the Navigator software integrated in the LAS X v 3.8.1 (Leica-Microsystems). CellProfiler 3.1.9 software (Broad Institute, https://cellprofiler.org/) was used for image analysis (nuclei identification and POLA1 signal intensity quantification).

## Alkaline comet assay

HCT116 cells synchronised in S phase were treated with VCPi (NMS 873). At the end of the treatment nascent DNA was labelled with a 1 h pulse of 30 μM EdU. Cells were harvested by trypsinization and

resuspended in cold PBS. 20 μL of a $10^6$ cells/mL suspension was mixed with 600 μL of 0.8% low melting point (LMP) agarose in PBS at 37 °C, and 60 μL of this cell-LMP suspension was spread onto slides coated with 1% LMP agarose. The slides were incubated for 20 min at 4 °C to allow the cell-agarose suspension to solidify and lysed overnight in homemade lysis solution (2.5 M NaCl, 100 mM EDTA, 10% DMSO, 1% Triton X-100, 10 mM Tris pH 10) or commercial lysis solution (Biotechne, 4250-050-01). Slides were then incubated in cold alkaline electrophoresis buffer (300 mM NaOH, 2.5 mM EDTA) for 40 min at 4 °C. Alkaline electrophoresis was performed for 20 min at 18 V/ 300 mA and at 4 °C. Next, slides were rinsed twice in cold PBS for 5 min and rinsed with water once before incubation in pre-cooled 70% ethanol for 10 min at room temperature. After drying the slides in an incubator at 37 °C, a click reaction was performed to label EdU with a 488 nm fluorophore-azide (AF488-Azide, Jena Bioscience, CLK-1275-1). The click reaction was performed in the presence of 10 mM sodium ascorbate, 2 mM $CuSO_4$ and 1 mM Azide-488 in PBS for 30 min at room temperature. Slides were washed twice with cold PBS for 10 min and rinsed briefly with water before drying in an incubator at 37 °C. Alternatively, the samples were stained with SYBR Gold (Invitrogen, S11494) to visualize total DNA. Microscopy was performed on an Axiovert200 Fluorescence Resonance Energy Transfer microscope (Zeiss), using a 20X magnification. Substract background was performed on the images. Images were analysed using the Open Comet plugin for Fiji[94].

## Statistical analysis

In DNA fiber experiments three independent replicates are shown. In scatter dot plots (fork rate, CldU/IdU ratio and long/short ratio) medians comparison was assessed using the non-parametric tests Mann-Whitney rank sum test (2 samples) or Kruskal-Wallis followed by Dunns multiple comparisons test (more than 2 samples); in bar graphs (1st label origin firing, terminations and percentage of asymmetric origins) results are expressed as mean ± SD and statistical analyses of sample means were performed with two-tailed Student's t tests (2 samples) or one-way ANOVA followed by Tukey's multiple comparisons test (more than 2 samples).

Immunofluorescence signal quantifications (Fig. 3D) were represented as a pool of three independent experiments. Median differences were assessed with the Mann–Whitney rank sum test.

For immunoblot quantifications Kruskal–Wallis (Fig. 4D and Supplementary Fig. 4C) or 2-way ANOVA (Supplementary Fig. 2G), were used to compare samples.

Comet assays were analyzed using Mann–Whitney except in Supplementary Fig. 7C, D were Kruskal–Wallis test was used. A minimum of three replicates are shown in the figures

Statistical analysis was performed in Prism v9.5 (GraphPad Software). Statistical significance: *$p$-value < 0.05; **$p$ < 0.01; ***$p$ < 0.001.

## Reporting summary

Further information on research design is available in the Nature Portfolio Reporting Summary linked to this article.

## Data availability

All data supporting the findings of this study are available within the paper and its Supplementary Information files. Source data are provided with this paper.

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

## Acknowledgements

Research was funded by grants from the Spanish Ministry of Science (grant BFU2014-55168-JIN, funded by MICIU/AEI and ERDF A way of making Europe; grant RTI2018-093485-B-I00, funded by MICIU/AEI and ERDF A way of making Europe; grant PID2021-127824OB-I00, funded by MICIU/AEI and ERDF/EU; grant CNS2022-135442, funded by MICIU/AEI and by European Union NextGenerationEU/PRTR) and a Ramón y Cajal Fellowship (grant RYC-2016-20705, funded by MICIU/AEI and ESF investing in your future) to Emilio Lecona; grant from the Spanish Ministry of Science (grant PID2019-106707-RB, funded by MICIU/AEI and ERDF A way of making Europe; grant PID2022-142771NB-I00 funded by MICIU/AEI and FEDER, UE) to Juan Méndez; and an ERC Starting Grant (ERC-StG-679754) and a grant from the Danish National Research Foundation, Denmark (DNRF115)a (BES-2015-075758) to Luis Toledo. We want to acknowledge the assistance of the flow cytometry and advanced microscopy facilities from the CBMSO. Scott B Churcher is part of the RepliFate Doctoral Network (HORIZON-MSCA-2021-DN-01-01 Project 101072903). We would like to thank the Penengo lab for their help with the EdU comet assays.

## Author contributions

S.R.A. and R.M.R. carried out most of the experiments in the study. A.G.M. carried the immunoprecipitation experiments and the purification of ubiquitylated proteins with the His-ubiquitin system. S.B.C. helped in the immunoprecipitation experiments. A.P. participated in immunoprecipitation experiments. A.F.L. and G.V.B. carried some experiments in Supplementary Figs. 3 and 5. PO carried out the experiment in Supplementary Fig. 2A. EMD carried out the experiment in Supplementary Fig. 4D. JM and LT contributed to experimental design and the revision of the manuscript. EL designed the experiments, coordinated the study, and wrote the manuscript.

## Competing interests

The authors declare no competing interests.
