## [Peer Review File · Nature Communications]

DNA polymerase α /Primase extraction from chromatin by VCP/p97 restricts ATR activation during unperturbed DNA replication

Corresponding Author: Dr Emilio Lecona

Version 0:

Reviewer comments:

Reviewer #1

(Remarks to the Author)

In this study, Rodríguez-Acebes et al. provide evidence that the basal activation of the ATR kinase correlates with the loading of the DNA pol α /primase (POLA/PRIM) complex on chromatin. The authors suggest that the ubiquitin-dependent segregase VCP/p97 extracts and, thereby, restricts the amount of POLA/PRIM on chromatin to limit the number of Okazaki fragments, consequently limiting the activation of ATR. Finally, they demonstrate that a VCP inhibitor triggers an increase in Okazaki fragments and ATR activation.

The proposed mechanism is conceptually interesting and the experiments are generally of high quality. Although often correlative, it would strengthen the manuscript to attempt to increase chromatin-bound POLA/PRIM levels in other ways than VCP inhibition, which triggers many processes at the same time (see point 7).

Another issue is the link between VCP and other POLA/PRIM subunits, as well as the phenotypic differences between knockdown of individual POLA/PRIM subunits (see points 2-6).

This study would deserve being considered for publication in Nature Communications, should the authors appropriately address the following points:

MAJOR POINTS

(1) Fig 2E, 'The formation of new forks is reduced when POLA is inhibited, leading to an increase in the speed of ongoing replication forks (Figure 2E-F and S2D)'.

How do the authors know that the formation of new forks is reduced? Which experiment shows this? Why is the speed of 'ongoing' forks not reduced when POLA is inhibited? Would this not be expected? Treatment of cells with POLAi CD437 slows replication fork speed, according to PMID: 35353580. Is this a difference in methodology or the specific POLAi used (ST1926 in this study)? This should be better explained or addressed experimentally.

(2) Fig S3B. 'Conversely, VCP/p97 pull-down recovered one of the specific bands in POLA1 (Figure S3B, marked with an asterisk), and this band is also pulled-down with a specific antibody against POLA1 (Figure S3C, marked with an asterisk).'

How sure are the authors that this is a specific interaction between POLA1 and VCP? In this case, VCP does not co-IP PRIM2 and only one band is detected for POLA1. It would help to show that this band is lost when the IP is done in POLA1 knock-down cells to demonstrate specificity of the band / antibody.

(3) Fig 3B. 'The immunoprecipitation of UFD1L confirmed an interaction with POLA1 as well as small amounts of PRIM2.'

While the PRIM2 co-IP convincingly shows an interaction with UFD1L, the reciprocal co-IP is not very convincing. The amount of POLA1 in the IgG control or the UFD1L co-IP are nearly equal. This should be improved, or the conclusion toned down.

(4) Fig 3B. 'Tandem Ubiquitin Binding Entities (TUBE) showed that ubiquitylated POLA1 is present in control conditions and accumulates in response to VCPi.'

The increase after VCPi is clear, but I do not really see ubiquitylated POLA1 in the control condition. Why do the authors only look at POLA1? Could other subunits of POLA/PRIM not be ubiquitylated and targeted by VCP? For example, Fig 3C shows no increase in chromatin-bound POLA1, while POLA2 and PRIM2 are increased. Would this not suggest that POLA2 is a more likely substrate? The authors should use the same TUBE approach to test other POLA/PRIM subunits.

(5) Fig 3C. 'We reasoned that the lack of accumulation of POLA1 is due to the increase in ubiquitylated POLA1 (Figure 3B).'

The amount of ubiquitylated POLA1 is very little (in Fig 3B) when compared to the unmodified protein pool. It is difficult to image that the increase in POLA2 is not accompanied by an increase in POLA1 on chromatin, if the latter is the substrate of the VCP segregase. Can the authors use another way (e.g. imaging, flow cytometry) to investigate this further?

(6) Fig 4A, B. 'CHK1 phosphorylation was increased upon VCPi treatment in cells transfected with a non-specific siRNA and this effect was attenuated when the levels of POLA or PRIM complexes were reduced.'

CHK1 phosphorylation shows varying effect when depleting different POLA or PRIM subunits. For instance, the levels of CHK1 phosphorylation are higher in POLA1-depleted cells, even at the lowest concentration of VCPi. By contrast, in POLA2-depleted cells, the levels of CHK1 phosphorylation are lower, even at the highest VCPi concentration. Also PRIM1 and PRIM2 depletion do not show the same phenotype, with PRIM2 depletion not having a marked impact on CHK1 phosphorylation in Fig 3B.

It is also not clear to me why POLA1/PRIM2 depletion triggers increased RPA2 phosphorylation, while POLA2/PRIM do not. As far as I can tell, this is not explained anywhere.

(7) VCP inhibition will affect many processes at the same time, leading to pleiotropic effects. The authors should try to attempt to increase the levels of chromatin-bound POLA/PRIM in other ways, to further validate their findings. One possible option would be to fuse the USP2 deubiquitylase to either POLA1 or POLA2, to restrict the levels of ubiquitylated

MINOR POINTS

(a) Fig 1H. Why not show the short/long ratio? In that way, a symmetrical fork is 1, while an asymmetrical fork is shorter than 1.

(b) Asymmetrical forks are often considered to be a consequence of R-loops. Any thoughts on the source of the asymmetry in this case?

(c) Fig S3B. Is it expected that less POLA1 is present in the co-IP with PRIM2 when cells are treated with POLA inhibitor?

STATISTICS

applaud showing all datapoints as much possible, as done in many panels, including for instance Fig 1D. However, data representation could be improved by adding the median/mean for each biological replicate as well. Now, all datapoints are 'piled up' without showing whether the phenotypes were similar or different between experiments. Showing all datapoints and the mean (as shown now), and in addition also showing the mean for each experiment, allows the reader to get a sense of how reproducible phenotypes are between experiments.

TEXTUAL

Page 4: '... protein through the central pore of the complex (22–24)). There is an additional 'b' after complex.

Reviewer #2

(Remarks to the Author)

In their manuscript entitled "The activation of ATR during unperturbed DNA replication is restricted by VCP/p97 through the extraction of DNA polymerase α /Primase from chromatin" Sara Rodríguez-Acebes, Rodrigo Martín-Rufo and coauthors describe an unappreciated role for VCP in the regulation of early events during DNA replication and how this process is incorporated into genome wide replication surveillance through the activation of the ATR/CHK1 checkpoint kinases, counteracting genotoxic DNA damage.

Throughout the manuscript the authors base their conclusions on DNA fiber analyses and western blotting to analyze replication fork dynamics and checkpoint activation, respectively, representing sensitive and reliable techniques. For most experiments synchronized cell cultures were used in order to analyze direct effects in the consecutive cell cycle after release from thymidine block, minimizing inherited side effects from previous rounds of DNA replication. Systematically combining inhibitors for distinct regulatory processes that control DNA replication fork pattern, the authors dissect the role of VCP activity to take place early after replication origin firing. Thus, the regulatory mechanism described in this manuscript, are not

related the role of VCP in DNA replication termination, a late event in the replication program, consequently eluting on an unappreciated regulatory mechanism. The authors hypothesize that the POLA/PRIM polymerase complex required for replication initiation and generation of Okazaki fragments is regulated in its chromatin abundance by VCP. Accordingly, they show retention of POLA/PRIM upon VCP inhibition, correlating with ATR/CHK1 checkpoint activation through TOPBP1. The authors conclude that excessive POLA/PRIM activity upon VCP inhibition generates surplus Okazaki fragments that serve as recruiting entity for TOPBP1 and consequently checkpoint activation. Checkpoint activation, in turn, is required to compensate aberrant replication patterns and limit occurrence of genotoxic DNA damage.

The key findings of the manuscript are of broad interest and well suit for publication with Nature Communications. The consequences of VCP inhibition on the early replication pattern are well elucidated and reveal a regulatory role of VCP in early DNA replication events. Furthermore, the manuscript provides an underappreciated perspective on ATR/CHK1 activation during the canonical DNA replication program. However, the proposed excess in Okazaki fragment generation and consequently checkpoint activation should be supported by further experimental evidence.

Major concerns:

1) The authors' conclusion that excessive Okazaki fragments emerge upon VCP inhibition should be further confirmed. If possible, they could use an alternative assay than the comet tail moment quantification as this is supposed to report nascent DNA products which could also be distinct from Okazaki fragments.

2) related to 1) In any case, the comet assay should be supported by controls. How does POLA inhibition affect Okazaki fragment abundance on its own and in combination with VCP inhibition? This experiment is crucial to interrogate a causative effect on Okazaki fragment generation. Moreover, in order to support that excessive Okazaki fragments activate ATR/CHK1, can the authors confirm this hypothesis in another context than VCP inhibition (e.g. FEN1 depletion, LIG1 depletion)?

3) The authors use western blotting and detection of phosphorylated CHK1 as readout for ATR/CHK1 checkpoint activation. In several panels it is somewhat hard to conclude about the relative phospho-CHK1 levels compared to control conditions, as the changes are sometimes subtle while variations are also given in the loading control(s). It would be very helpful to show relative quantification of these western blots, ideally including replicates for central experiments (Figures 2K, 3D, 4A/B). The same applies to the chromatin retention analyses (Figures 3C, 3D). Quantification plots will facilitate conclusive interpretation and support robustness of the proposed model.

Minor concern/comments

4) The authors should be consistent in how high content data is presented. In particular regarding fork rate, CldU/IldU, and long/short fork plots. In the main figures mostly single experiments with high statistical power are shown (due to high amount of data points per condition). In the supplementary they present the cumulative mean data of the replicates. In most of the cases, the cumulative data confirms the authors conclusions, despite the statistical power is (as to be expected due to lesser amount of data points) lower. Nonetheless, the cumulative data is more reliably revealing reproducible effects and should, in our opinion, be shown in the main figures.

We note that e.g. the EdU tail moment quantification did not reveal statistical significant differences upon VCP inhibition when plotting the mean (Figure S6C) while the single experiment data in 5F is indicated as highly significantly different to the control. Central experiments should be supported by solid statistical analyses.

5) Figures 2/S2 and 4/S4 show epistasis analyses of VCP inhibition with either POLA1 inhibition or POLA/PRIM depletion. Based on the model that VCP primarily affects the chromatin abundance of POLA/PRIM, both experimental approaches would have been expected to result in slightly distinct outcome, which the authors conclude for the checkpoint activation aspect (page 12 of the manuscript). VCP inhibition and POLA1 inhibition should rather synergize, due to increased levels of non-productive POLA. In contrast POLA/PRIM depletion would have been expected to reduce the effect of VCP inhibition. It remains unclear, however, why both experimental approaches show the same effect on replication fork properties. The authors argue that the effect of VCP/POLA inhibition on the replication pattern defines checkpoint activation. If the replication pattern is comparable, how do the authors explain differential checkpoint activation upon POLA1 inhibition vs. POLA/PRIM depletion? An epistasis analysis using the comet tail moment readout might be helpful to address this question.

6) According to the authors' model, excess Okazaki fragments serve as recruiting signal for TOPBP1 and consequently ATR/CHK1 activation. As checkpoint activation is expected to affect replication fork progression and origin firing, the authors should carefully discuss what is primary consequence and what is a secondary effect of ATR/CHK1 activation on replication fork properties. Technically, it would be feasible to combine VCP/POLA inhibition with ATR/CHK1 inhibition to experimentally validate this aspect.

7) The representation of DNA fiber patterns in Figure 1C is somewhat confusing. In the upper row, the pulse labelling strategy is shown, while the lower panel depicts the labelling pattern that served as the readout for interpretation. This should be labelled clearly, to avoid confusion of both pictograms. It also remains vague in how far the determined properties of replications forks reflects specific defects in replication regulation, e.g. what does the CldU/IldU ratio report, what is the concept behind fork asymmetry? The authors should define how they interpret different fork properties.

8) Figure 3B shows an isolation experiment of POLA1 from control and VCP inhibited cell lysates using TUBEs. The fact that the higher molecular weight bands of POLA1 are almost identical in the two input conditions along with the fact that in both conditions mainly unmodified POLA1 was enriched, leaves a doubt about the specificity of a) the TUBE enrichment of ubiquitin conjugates and b) the POLA1 detection. This experiment should be interpreted with caution.

Reviewer #3

(Remarks to the Author)

The manuscript by Rodriguez-Acebes et al, "The activation of ATR during unperturbed DNA replication is restricted by

VCP/p97 through the extraction of DNA polymerase α /Primase from chromatin”, aims to address the role of VCP during DNA replication, beyond its known role in termination and prevention of re-replication. Here, data show that VCP inhibition causes several phenotypes: (i) a mild decrease in the progression of forks, and an increase in asymmetric forks (indicative of fork stalling). (ii) A strong decrease in origin firing and reduced EdU incorporation. (iii) A mild but reproducible activation of checkpoint signalling, which causes G2/M arrest, and which can be rescued by inhibition of CHK1 kinase. Authors suggest that VCP functions in regulating primer synthesis on the lagging strand, following fork activation. The proposed model suggests that VCP removes DNA polymerase α /Primase from chromatin during unchallenged replication, thereby limiting the number of Okazaki fragments and reducing the basal levels of checkpoint signalling. In the absence of VCP, there is over-priming of the DNA, excessive TOPBP1 recruitment and checkpoint signalling. This is an interesting idea, as little is understood about how VCP functions in DNA replication, aside from CDT1 degradation and replisome disassembly, but unfortunately there is currently not enough strong evidence to support the proposed model. Evidence that VCP and Pol α /PRIM work in the same pathway stems from the fact that inhibition of POLA and VCP, individually or together, causes the same level of fork stalling (FIG 2G), decreased origin firing (FIG 2F) and both prevent the increase in fork speeds caused by CDC7i. The checkpoint signalling seen in the VCPi-treated cells is also certainly replication-dependent, because co-inhibition of CDC7 (FIG 3D) and co-depletion of TOPBP1 (FIG S3F), which are known to promote fork activation, lead to convincing reductions in CHK1 phosphorylation. Many elements of the proposed model are however not supported strongly enough by the experimental data. This is especially the case for the direct role of VCP removal of POLA/primase from chromatin. While this work is interesting and worthwhile investigating, it requires more work to support the proposed model before being ready for publication in Nature Communications.

Major comments

1. The molecular weight markers are missing on all blots throughout the paper and it is difficult to judge which of the multiple bands is of a correct size and whether ubiquitylated forms are mono or polyubiquitylated. It is of importance as VCP shows a threshold of 5 ubiquitins added for processing.
2. Many of the accumulation or checkpoint activation effects observed by western blotting are rather mild and raise questions of reproducibility and robustness. Quantification of a number of experiments would increase the confidence in the proposed conclusion.
3. Authors need to address whether any of the VCPi phenotypes they observe could be explained by the known role of VCP in degrading CDT1 and the subsequent re-replication that would ensue. This role is mentioned in introduction but ignored afterwards.
4. Figure 3B lacks necessary controls – currently it looks like there is more ubiquitylated POLA in input than in TUBE pulldown. Is there enrichment of any ubiquitylated products in TUBE pulldown over nonspecific beads? Is there enrichment of ubiquitylated POLA in TUBE pulldown over nonspecific beads? Is this reproducible? If VCP is required for continuous removal of excess POLA/primase from unchallenged replication forks then one can expect this interaction to be more robust (especially after inhibition of p97 activity), but very little interaction is observed (Fig 3 and S3). Moreover, one would expect interaction of VCP with polyubiquitylated forms of POLA, while authors focus on non-modified POLA. Similarly, one would expect accumulation of ubiquitylated forms of POLA on chromatin (and figure 3B suggest that these can be visible in input), but only unmodified forms are presented in Figure 3C. Moreover, the conclusion that VCPi does not affect retainment of Cdc45 and PCNA on chromatin is misleading as the provided timecourse does not cover whole replication reaction – the level of PCNA and Cdc45 does not change over the timecourse, suggesting that cells do not reach end of the S-phase.
5. TOPBP1 is needed for checkpoint activation but also for initiation of DNA replication. Upon downregulation of TOPBP1 (Fig S3F) it is likely that observed effects are due to inhibition of origin firing (akin CDC7i) rather than loading of TOPBP1 on primers in order to activate checkpoint. There is many controls required to draw the conclusion proposed.
6. The presented experiments with downregulation of different components of POLA/primase complex are very confusing as showing very different phenotypes for downregulation of different subunits. If the major function of these subunits is to fulfil POLA/primase role as a part of the complex, one would expect them to have similar phenotypes. As it stands these differences suggest off-target effects of different siRNAs used, and suggest that authors should rescue the observed phenotypes using siRNA-resistant re-expression of proteins to confirm that these are not off-target artefacts.
7. If VCPi reduces priming while also reducing general origin firing and DNA synthesis, one would expect not more Okazaki fragments after VCPi, but shorter Okazaki fragments. Authors should use alkaline gel electrophoresis to measure the size of Okazaki fragments more precisely and also compare VCPi-treated cells with POLAi-treated cells and be able to rescue the phenotype.
8. Finally, in order to confirm this model further, authors should identify the VCP cofactor(s) implicated in this mechanism as this would help to relieve some of the pleiotropic effects of VCP inhibition and make the data much more robust.

Minor comments

1. What are the units the tail moment is quantified in Figure 5F and Figure S6?
2. It would be helpful to explain at the beginning of the study the mode of action of POLAi and concentration required to inhibit its activity. Only towards the end of the manuscript it is revealed that the concentration used throughout does not strongly block POLA activity. The direct measure of the level of inhibition would also help to support the conclusions drawn.

Version 1:

Reviewer comments:

Reviewer #1

(Remarks to the Author)

The authors have carefully addressed all my points and have significantly improved the manuscript.

1. The new interaction data in Fig 3A (together with knockdown control in Fig S3B) is a big improvement and strengthens the conclusions considerably.
2. The His-Ubiquitin approach in Fig S3C is also much better than the previous TUBEs approach. If the authors have (negative) data on POLA2 and PRIM1 using this approach, it would be worthwhile to include this too.
3. The new POLA1 imaging data in Fig 3D, and the combined POL/PRIM knockdown in Fig 4C/Fig S4A is much more convincing now.

Reviewer #2

(Remarks to the Author)

The authors have convincingly addressed all the issues raised.

The manuscript is a strong contribution to the broad readership of Nature comms.

Reviewer #3

(Remarks to the Author)

The revised version of the manuscript by Rodriguez-Acebes et al. has improved and thank you for addressing a number of my and other reviewers comments. Thank you for the markers, quantifications etc.

I still believe that this is an interesting problem, but still have a number of questions about the key experimental findings of the manuscript.

The authors removed some of the problematic experiments and provided new ones, but the new ones lead to new questions and are missing controls.

The main part that is not convincing is the data supporting the main hypothesis that VCP/p97 removes ubiquitylated POL1/PRIM from chromatin.

This is based on:

1. - interaction of POL1/PRIM2 with VCP/UFD1 – this is now mainly based on UFD1 immunoprecipitation (Figure 3B), however in this IP UFD1 does not interact with VCP/p97? One would expect UFD1 to interact robustly with VCP/p97 – is this then a subcomplex that does not work with VCP/p97?
2. - retention of POLA1/PRIM2 on chromatin in VCPi – this is not visible on chromatin by western blot and authors argue that this is due to inability to visualise ubiquitylated forms of these proteins – what about treating samples with DUBs that would cut off the chains and visualise accumulated unmodified proteins? Second way of visualising POLA1/PRIM2 on chromatin that authors provided now is IF. However, the experiments provided have no controls. Could the authors provide negative controls for specificity of antibodies – e.g. siRNA of PRIM2/POLA1 to show that signal disappears, look at signal at different cell cycle stages to get a base line in G1, increase in S-phase and then even more accumulation in VCPi, positive control with some known treatment that increases POL alpha on chromatin. This is a key experiment and needs controls.
3. – accumulation of ubiquitylated forms of PRIM2/POLA1 in VCPi treatment – authors here exchange tube experiment for His-Ubi pulldown, but the new experiment although showing showing robust ubiquitin pulldown, presents very minimal levels of POLA1/PRIM2 signal.

Altogether, with the three main three lines of evidence having problems, I am still not convinced that POLA1/PRIM2 is a VCP target.

I appreciate also that alkaline gel electrophoresis is a new technique to be set up, however I believe that it provides much more information than tail moment analyses provided. Tail moment measured can be effect of any fragments of DNA containing nascent DNA and does not allow to distinguish the differences in the length and abundance of the Okazaki fragments. To convincingly show that VCPi affects priming more detailed analysis is needed. For example – the low concentration of POLAi induces inhibition of new origin firing and change of fork speed (Figure 2) but not changes in tail moment, interpreted as no changes in priming (Figure 6). Where does the effect observed in fibres come from then, if not from differences in priming?

Finally, two of the reviewers highlighted the pleiotropic effects of VCPi and the need to narrow down the cofactors responsible, to have a parallel path of validation of the mechanism beyond the pleiotropic VCP. There is a handful of known cofactors working with VCP on chromatin and it is not beyond the scope to siRNA them and see which one reproduces the effects on POL alpha observed with VCPi.

Altogether, though I believe that the manuscript is improved, in my opinion more work is needed and more controls provided to conclusively say that POLA1/PRIM2 in VCP substrate and that VCP regulates priming activity.

Version 2:

Reviewer comments:

Reviewer #3

(Remarks to the Author)

I would like to thank the authors for extensive response to my comments in the authors' rebuttal. I am a little disappointed that not much of what is shown in the rebuttal has made it into the manuscript itself.

Thank you for showing me that the immunofluorescent antibody against PolA1 is specific, however, I believe that this is an essential control that should be included within the manuscript, too. Thank you also for removing the staining with PRIM2 that turned out not to be specific.

Thank you for the new version of the HIS-Ubi pull-down – indeed, it does look better.

The authors showed just in rebuttal a different version of the PRIM2 and UFD1L immunoprecipitation, that indeed shows some VCP interacting with UFD1L. I can see how the authors may not wish to repeat the same blot within manuscript, but maybe they could include quantification of the IP data to indicate reproducibility of their results over a number of experiments?

The authors included also in rebuttal data on downregulation of one of the major cofactors of VCP (NPLOC4) and one of the minor cofactors (FAF2), which lead to small accumulation of POLA/PRIM. These data does strengthen the main message of this manuscript – that VCP controls the level of chromatin bound POLA/PRIM, and also addresses the point raised by two of the reviewers about the pleiotropic nature of inhibition of all of the VCP activity when using the inhibitor. I strongly believe that these data should be included in the manuscript to strengthen it. The authors present these data as a response to 3 of my points, but do not include them in the manuscript.

Finally, my understanding of the description of the chosen low concentration of POLAi inhibitor, here and in cited previous papers, is that the moderate accumulation of ssDNA and RPA on chromatin, and inhibition of origin firing, results from low level of inhibition of priming activity provided by POLA/PRIM. However, if the authors believe that there is no effect on priming, based on their EdU and DNA tail moment (manuscript and rebuttal), then this is an important message to introduce as a concept when introducing the experimental setting on page 7. This will disperse the assumption that POLAi inhibitor inhibits the priming activity at all concentrations that show phenotypic changes.

Overall, I believe that the indicated above information that was provided just for reviewers is important to include within the manuscript.

Thank you for addressing my comments.

REVIEWER COMMENTS

Reviewer #1 (Remarks to the Author):

In this study, Rodríguez-Acebes et al. provide evidence that the basal activation of the ATR kinase correlates with the loading of the DNA pol α /primase (POLA/PRIM) complex on chromatin. The authors suggest that the ubiquitin-dependent segregase VCP/p97 extracts and, thereby, restricts the amount of POLA/PRIM on chromatin to limit the number of Okazaki fragments, consequently limiting the activation of ATR. Finally, they demonstrate that a VCP inhibitor triggers an increase in Okazaki fragments and ATR activation.

The proposed mechanisms is conceptually interesting and the experiments are generally of high quality.

We thank the reviewer for the appreciation of the quality of our work and its impact.

Although often correlative, it would strengthen the manuscript to attempt to increase chromatin-bound POLA/PRIM levels in other ways than VCP inhibition, which triggers many processes at the same time (see point 7).

Another issue is the link between VCP and other POLA/PRIM subunits, as well as the phenotypic differences between knockdown of individual POLA/PRIM subunits (see points 2-6).

This study would deserve being considered for publication in Nature Communications, should the authors appropriately address the following points:

MAJOR POINTS

(1) Fig 2E, ‘The formation of new forks is reduced when POLA is inhibited, leading to an increase in the speed of ongoing replication forks (Figure 2E-F and S2D)’.

How do the authors know that the formation of new forks is reduced? Which experiment shows this?

DNA fiber assays measure fork rate by directly evaluating the length of the second label (Figure 2E) and the firing of new origins by following the appearance of green tracks flanked by red tracks in both directions (Figure 2F). The formation of new forks requires the firing of new origins. Thus, measuring origin firing also reflects the formation of new DNA replication forks.

Why is the speed of ‘ongoing’ forks not reduced when POLA is inhibited? Would this not be expected? Treatment of cells with POLAi CD437 slows replication fork speed, according to PMID: 35353580. Is this a difference in methodology or the specific POLAi used (ST1926 in this study)? This should be better explained or addressed experimentally.

In these experiments we used a concentration of POLAi (adarotene-ST1926) that partially inhibits the activity of POLA1. We chose this concentration since it does not lead to the toxic accumulation of ssDNA and the induction of replication catastrophe leading to DNA damage (Ercilla et al., Cell Rep 2020). We have changed the text to clarify this point earlier in the manuscript since it was confusing as indicated by Reviewer 3. ST1926 and CD347 share the same chemical backbone and are expected to have similar effects though with different potency. In their 2022 paper Mehta *et al.* used a much higher concentration of POLA1 inhibitor that should completely block the catalytic activity of POLA1. Accordingly, we have found that increasing the concentration of Adarotene/ST1926 reduces fork rate in HCT116 cells (Reviewer Figure 1, left) as the result of the almost complete inhibition of DNA replication (Reviewer Figure 1, right).

Reviewer Figure 1. Effects of increasing concentrations of POLA1 inhibitors on DNA replication.

HCT116 cells were synchronized in G1/S in the presence of 2 mM thymidine for 16 h. Cells were released for 2 h and then treated with DMSO (Control) or increasing concentrations of Adarotene/ST1926 for 2 h. Fork rate (left) was measured in DNA fiber assays (3 independent experiments were performed) and DNA replication (right) was measured by flow cytometry, following total DNA content with DAPI and DNA replication with EdU labelling.

(2) Fig S3B. 'Conversely, VCP/p97 pull-down recovered one of the specific bands in POLA1 (Figure S3B, marked with an asterisk), and this band is also pulled-down with a specific antibody against POLA1 (Figure S3C, marked with an asterisk).'

How sure are the authors that this is a specific interaction between POLA1 and VCP? In this case, VCP does not co-IP PRIM2 and only one band is detected for POLA1. It would help to show that this band is lost when the IP is done in POLA1 knock-down cells to demonstrate specificity of the band / antibody.

As suggested, we have performed VCP/p97 pull-downs after POLA1 depletion. Under these conditions a band is still detected with the POLA1 antibody in the VCP/p97 pull-down, indicating this is probably a non-specific band. Thus, we have removed this experiment. To confirm the interaction of VCP/p97 with POLA/PRIM we have improved our pull-down experiments with UFD1L including additional experiments that further substantiate the interaction between POLA/PRIM and VCP/p97-UFD1L (see next point).

(3) Fig 3B. 'The immunoprecipitation of UFD1L confirmed an interaction with POLA1 as well as small amounts of PRIM2.'

While the PRIM2 co-IP convincingly shows an interaction with UFD1L, the reciprocal co-IP is not very convincing. The amount of POLA1 in the IgG control or the UFD1L co-IP are nearly equal. This should be improved, or the conclusion toned down.

We have repeated the pull-down experiments. Figure 3A now shows clearly that pulling-down PRIM2 recovers POLA1 as well as VCP/p97 and UFD1L. Conversely, Figure 3B shows that UFD1L pull-down recovers POLA1, PRIM2 and VCP/p97. In addition, we have carried out pull-down experiments after POLA1 depletion (Figure S3B). As expected, the loss of POLA1 is clearly observed in the UFD1L pull-down. In addition, the binding of PRIM2 is also lost in the absence of POLA1 even if the levels of PRIM2 remain unaffected in the input. These results show that UFD1L recovers the whole POLA/PRIM complex and that POLA1 is necessary for this interaction.

(4) Fig 3B. 'Tandem Ubiquitin Binding Entities (TUBE) showed that ubiquitylated POLA1 is present in control conditions and accumulates in response to VCPi.'

The increase after VCPi is clear, but I do not really see ubiquitylated POLA1 in the control condition. Why do the authors only look at POLA1? Could other subunits of POLA/PRIM not be ubiquitylated and targeted by VCP? For example, Fig 3C shows no increase in chromatin-bound POLA1, while POLA2 and PRIM2 are increased. Would this not suggest that POLA2 is a more likely substrate? The authors should use the same TUBE approach to test other POLA/PRIM subunits.

After repeating the TUBE experiments we did not manage to get consistent clean results with this system. As an alternative approach we transfected HCT116 cells with His-tagged ubiquitin and pulled-down ubiquitylated proteins with a Ni-NTA agarose resin both in control treated cells and upon inhibition of VCP/p97. Figure S3C shows the data with this system where we detect POLA1 and PRIM2 ubiquitylation, with a slight accumulation in VCPi treated cells. We did not detect POLA2 and PRIM1 ubiquitylation.

(5) Fig 3C. 'We reasoned that the lack of accumulation of POLA1 is due to the increase in ubiquitylated POLA1 (Figure 3B).'

The amount of ubiquitylated POLA1 is very little (in Fig 3B) when compared to the unmodified protein pool. It is difficult to image that the increase in POLA2 is not accompanied by an increase in POLA1 on chromatin, if the latter is the substrate of the VCP segregase. Can the authors use another way (e.g. imaging, flow cytometry) to investigate this further?

We have carried out immunofluorescence experiments in pre-extraction conditions to confirm if POLA1 accumulates in chromatin upon VCP/p97 inhibition. In this setting we should be able to detect both ubiquitylated and non-modified POLA1. Figure 3D-E shows that both POLA1 and PRIM2 accumulate on chromatin upon

VCP/p97 inhibition. In addition, our pull-down experiments indicate that POLA1 is necessary for the interaction of UFD1L with the POLA/PRIM complex (Figure S3B). Together, these data support a role for POLA1 ubiquitylation in the recognition of the complex by VCP/p97. Further, our data show that POLA1 and PRIM2 are both ubiquitylated (Figure S3C). Further characterization of the ubiquitylation of the POLA/PRIM complex will be the subject of future experiments and is out of the scope of the current work.

(6) Fig 4A, B. 'CHK1 phosphorylation was increased upon VCPi treatment in cells transfected with a non-specific siRNA and this effect was attenuated when the levels of POLA or PRIM complexes were reduced.'

CHK1 phosphorylation shows varying effect when depleting different POLA or PRIM subunits. For instance, the levels of CHK1 phosphorylation are higher in POLA1-depleted cells, even at the lowest concentration of VCPi. By contrast, in POLA2-depleted cells, the levels of CHK1 phosphorylation are lower, even at the highest VCPi concentration. Also PRIM1 and PRIM2 depletion do not show the same phenotype, with PRIM2 depletion not having a marked impact on CHK1 phosphorylation in Fig 3B.

It is also not clear to me why POLA1/PRIM2 depletion triggers increased RPA2 phosphorylation, while POLA2/PRIM do not. As far as I can tell, this is not explained anywhere.

We think the variability in the effects of individual siRNA against the subunits of the POLA/PRIM complex is due to the levels of depletion achieved in each individual knockdown for the different components of the complex. To circumvent this issue, we combined the 4 siRNA directed against each of the subunits to deplete the whole complex at once. Using a lower concentration of each siRNA also precludes off-target effects. Under these conditions we observe a very strong depletion of POLA1, PRIM1 and PRIM2, with a milder effect on POLA2 (Figure S4A). As expected, the depletion of the POLA/PRIM complex led to increased phosphorylation of CHK1, RPA2 and γ H2AX (Figure 4C), reflecting the accumulation of ssDNA that results from the decreased priming detected in the EdU comet assay (Figure 6C-D).

Still, the treatment with VCP/p97 inhibitor did not increase the phosphorylation of CHK1 in POLA/PRIM depleted cells, as can be observed in Figure 4C and in the quantification from 5 independent experiments shown in Figure 4D. VCP/p97 inhibition reduces fork speed (Figure 4A) and ssDNA accumulation (less RPA2 phosphorylation (Figure 4C) in POLA/PRIM depleted cells, which should reduce the levels of CHK1 phosphorylation. However, VCPi treatment also increases priming in POLA/PRIM depleted cells (Figure 6I-J), which should increase the levels of CHK1 phosphorylation. As a result, we observe no change in CHK1 phosphorylation upon treatment of POLA/PRIM depleted cells with VCPi (Figure 4C).

(7) VCP inhibition will affect many processes at the same time, leading to pleiotropic effects. The authors should try to attempt to increase the levels of chromatin-bound POLA/PRIM in other ways, to further validate their findings. One possible option would be to fuse the USP2 deubiquitylase to either POLA1 or POLA2, to restrict the levels of ubiquitylated

We appreciate that this is a very interesting point raised by the reviewer. Our experimental approach already minimizes the secondary effects of VCP/p97 inhibition by: i) working with cells synchronized in S phase to focus on DNA replication; and ii) limiting the treatment with the inhibitor to 2 h, a time when we do not see a general effect on the replication machinery. In fact, the accumulation of POLA/PRIM on chromatin can be observed even at shorter times by chromatin fractionation (Figure 3C). Together with the data showing a direct interaction of UFD1L with POLA/PRIM and the results with the His-Ub that show a slight accumulation of ubiquitylated POLA1 and PRIM2 after VCP/p97 inhibition, we think that an indirect effect of VCP/p97 on the accumulation of POLA/PRIM on chromatin is highly unlikely. Our future work will focus on the identification of the specific cofactors of VCP/p97 that target POLA/PRIM as well as in the identification of the ubiquitin ligase and deubiquitylase that modify the POLA/PRIM components. We think these experiments are interesting but lie beyond the scope of the current manuscript.

MINOR POINTS

(a) Fig 1H. Why not show the short/long ratio? In that way, a symmetrical fork is 1, while an asymmetrical fork is shorter than 1.

Using short/long or long/short ratio does not change the result. We have decided to use the long/short ration since it is the most commonly used representation of this kind of data in the field. In this way an increase in asymmetry is reflected by an increase in the ratio, which we feel enables a straightforward interpretation of the data.

(b) Asymmetrical forks are often considered to be a consequence of R-loops. Any thoughts on the source of the asymmetry in this case?

Every insult that blocks the progression of the replicative DNA polymerases will generate asymmetrical forks, including the accumulation of R-loops. In this case we favour a more direct mechanism where VCP/p97 inhibition impacts the progression of the lagging strand by affecting the priming activity of POLA/PRIM. Thus, we propose that asymmetry arises from the impaired progression of the lagging strand when VCP/p97 is blocked.

(c) Fig S3B. Is it expected that less POLA1 is present in the co-IP with PRIM2 when cells are treated with POLA inhibitor?

We think that the reduction in POLA1 levels in the IP after POLA inhibition are just due to experimental variability. In any case, we have included new immunoprecipitation experiments using UFD1L and PRIM2 after treatment with VCP/p97 inhibitor and we have decided to remove this data which was redundant.

STATISTICS

applaud showing all datapoints as much possible, as done in many panels, including for instance Fig 1D. However, data representation could be improved by adding the median/mean for each biological replicate as well. Now, all datapoints all 'piled up' without showing whether the phenotypes were similar or different between experiments. Showing all datapoints and the mean (as shown now), and in addition also showing the mean for each experiments, allows the reader to get a sense of how reproducible phenotypes are between experiments.

The mean or median of each experiment is now shown in the main figures and has been removed from the supplementary information.

TEXTUAL

Page 4: '... protein through the central pore of the complex (22–24)). There is an additional 'b' after complex.

The typo has been corrected.

Reviewer #2 (Remarks to the Author):

In their manuscript entitled "The activation of ATR during unperturbed DNA replication is restricted by VCP/p97 through the extraction of DNA polymerase α /Primase from chromatin" Sara Rodríguez-Acebes, Rodrigo Martín-Rufo and coauthors describe an unappreciated role for VCP in the regulation of early events during DNA replication and how this process is incorporated into genome wide replication surveillance through the activation of the ATR/CHK1 checkpoint kinases, counteracting genotoxic DNA damage.

Throughout the manuscript the authors base their conclusions on DNA fiber analyses and western blotting to analyze replication fork dynamics and checkpoint activation, respectively, representing sensitive and reliable techniques. For most experiments synchronized cell cultures were used in order to analyze direct effects in the consecutive cell cycle after release from thymidine block, minimizing inherited side effects from previous rounds of DNA replication. Systematically combining inhibitors for distinct regulatory processes that control DNA replication fork pattern, the authors dissect the role of VCP activity to take place early after replication origin firing. Thus, the regulatory mechanism described in this manuscript, are not related the role of VCP in DNA replication termination, a late event in the replication program, consequently eluting on an unappreciated regulatory mechanism. The authors hypothesize that the POLA/PRIM polymerase complex required for replication initiation and generation of Okazaki fragments is regulated in its chromatin abundance by VCP. Accordingly, they show retention of POLA/PRIM upon VCP inhibition, correlating with ATR/CHK1 checkpoint activation through TOPBP1. The authors conclude that excessive POLA/PRIM activity upon VCP inhibition generates surplus Okazaki fragments that serve as recruiting entity for TOPBP1 and consequently checkpoint activation. Checkpoint activation, in turn, is required to compensate aberrant replication patterns and limit occurrence of genotoxic DNA damage.

The key findings of the manuscript are of broad interest and well suit for publication with Nature Communications. The consequences of VCP inhibition on the early replication pattern are well elucidated and reveal a regulatory role of VCP in early DNA replication events. Furthermore, the manuscript provides an underappreciated perspective on ATR/CHK1 activation during the canonical DNA replication program. However, the proposed excess in Okazaki fragment generation and consequently checkpoint activation should be supported by further experimental evidence.

We thank the reviewer for the thoughtful appreciation of the relevance and consistency of our work. In the revised version we have included additional data to further support our proposed model, expanding the results with the EdU comet assay.

Major concerns:

1) The authors' conclusion that excessive Okazaki fragments emerge upon VCP inhibition should be further confirmed. If possible, they could use an alternative assay than the comet tail moment quantification as this is supposed to report nascent DNA products which could also be distinct from Okazaki fragments.

2) related to 1) In any case, the comet assay should be supported by controls. How does POLA inhibition affect Okazaki fragment abundance on its own and in combination with VCP inhibition? This experiment is crucial to interrogate a causative effect on Okazaki fragment generation.

As suggested by the reviewer, we conducted additional experiments to validate the use of the EdU comet assay to measure the priming activity of POLA/PRIM. We show that the tail moment for EdU is not changed upon partial inhibition of POLA1 (Figure 6A-B), indicating that this concentration of POLA1 only delays priming. In contrast, the EdU tail moment is significantly reduced by the depletion of the POLA/PRIM complex (Figure 6C-D), in line with the distributive model for priming proposed by Porcella *et al.* (PLoS Genetics 2020). The complete inhibition of POLA1 induces widespread fragmentation in the DNA that leads to an increase in the total DNA (SYBR) and the EdU tail moments (Figure S6C-D) due to the induction of replication catastrophe. In contrast, the partial inhibition of POLA1 or the depletion of POLA/PRIM did not change the tail moment of total DNA (Figure S6A-B). These results show that the EdU tail moment reflects the priming activity of POLA/PRIM in the absence of damage. This priming activity includes the generation of Okazaki fragments as well as the re-priming events in the lagging strand, also mediated by POLA/PRIM (Conti *et al.*, Nat Comm. 2024; Machacova *et al.*, Nat Comm 2024; Kolinjivadi *et al.*, Mol Cell 2017). It was previously shown that the EdU comet tail is lost upon the maturation of the fragments generated by POLA/PRIM in the absence of damage (Stoy *et al.* NSMB 2023).

We evaluated the effect of VCPi treatment on priming and found that blocking VCP/p97 increased the EdU tail moment by itself (Figure 6E-F) and also in combination with partial POLA1 inhibition (Figure 6G-H) and POLA/PRIM depletion (Figure 6I-J). As a control we showed that the treatment with VCPi did not significantly increase the total DNA tail moment (Figure S6E-G). We did not measure the effect in combination with the full inhibition of POLA1 since the onset of replication catastrophe precludes a clear analysis of the results. We conclude that VCP/p97 inhibition promotes the priming activity of POLA/PRIM even in the presence of reduced amounts of the complex.

The implementation of an alternative method to directly measure the Okazaki fragments would require a substantial amount of time and work. Instead, we decided to focus our efforts in the validation of the EdU comet assay to measure the priming activity of POLA/PRIM and its application to explore the functions of VCP/p97.

Moreover, in order to support that excessive Okazaki fragments activate ATR/CHK1, can the authors confirm this hypothesis in another context than VCP inhibition (e.g. FEN1 depletion, LIG1 depletion)?

This is a very interesting point raised by the reviewer. Defects in the maturation of Okazaki fragments have been previously shown to activate the RSR in other systems. A summary of the different studies can be found in the recent review by Sun *et al.* (Trends in Cell Biology, 2023). Further, the inhibition of lagging strand maturation has been linked to sensitivity to PARP inhibitors in cancer (Li and Zou, JCI 2024). In this context the development of resistance to PARP inhibitors requires the activation of ATR, again linking the maturation of Okazaki fragments with the RSR (Li and Zou, JCI 2024). These works have already shown that non matured Okazaki fragments do activate the RSR and we have included this point as part of the discussion in the manuscript.

3) The authors use western blotting and detection of phosphorylated CHK1 as readout for ATR/CHK1 checkpoint activation. In several panels it is somewhat hard to conclude about the relative phospho-CHK1 levels compared to control conditions, as the changes are sometimes subtle while variations are also given in the loading control(s). It would be very helpful to show relative quantification of these western blots, ideally including replicates for central experiments (Figures 2K, 3D, 4A/B). The same applies to the chromatin retention analyses (Figures 3C, 3D). Quantification plots will facilitate conclusive interpretation and support robustness of the proposed model.

We have included the quantification of Figure 2K in Figure S2G (two independent experiments quantified). The quantification of former Figure 3D (Figure 3F) is included in Figure S3F, showing 7 independent experiments. Former Figure 4A-B has been substituted with the combined knockdown of all the POLA/PRIM subunits in Figure 4C and S4A. The quantification from 5 independent experiments is included in Figure 4D.

Regarding the chromatin retention in Figures 3C-D we decided to carry out immunofluorescence analysis of chromatin bound POLA1 and PRIM2 using pre-extraction protocols that remove the soluble nuclear material. As shown in Figure 3D-E both POLA1 and PRIM2 strongly accumulate on chromatin after the inhibition of VCP/p97.

Minor concern/comments

4) **The authors should be consistent in how high content data is presented. In particular regarding fork rate, CldU/IdU, and long/short fork plots.** In the main figures mostly single experiments with high statistical power are shown (due to high amount of data points per condition). In the supplementary they present the cumulative mean data of the replicates. In most of the cases, the cumulative data confirms the authors conclusions, despite the statistical power is (as to be expected due to lesser amount of data points) lower. **Nonetheless, the cumulative data is more reliably revealing reproducible effects and should, in our opinion, be shown in the main figures.**

We note that e.g. the EdU tail moment quantification did not reveal statistical significant differences upon VCP inhibition when plotting the mean (Figure S6C) while the single experiment data in 5F is indicated as highly significantly different to the control. Central experiments should be supported by solid statistical analyses.

As requested by the reviewer we have included the cumulative mean data in the main figures, keeping the paired information of individual experiments with different symbols for each replicate. We have included the statistical analysis in all the DNA fiber and EdU comet assay experiments.

5) **Figures 2/S2 and 4/S4 show epistasis analyses of VCP inhibition with either POLA1 inhibition or POLA/PRIM depletion.** Based on the model that VCP primarily affects the chromatin abundance of POLA/PRIM, **both experimental approaches would have been expected to result in slightly distinct outcome**, which the authors conclude for the checkpoint activation aspect (page 12 of the manuscript). VCP inhibition and POLA1 inhibition should rather synergize, due to increased levels of non-productive POLA. In contrast POLA/PRIM depletion would have been expected to reduce the effect of VCP inhibition. It remains unclear, however, why both experimental approaches show the same effect on replication fork properties. The authors argue that the effect of VCP/POLA inhibition on the replication pattern defines checkpoint activation. **If the replication pattern is comparable, how do the authors explain differential checkpoint activation upon POLA1 inhibition vs. POLA/PRIM depletion?** An epistasis analysis using the comet tail moment readout might be helpful to address this question.

To understand the phenotypes elicited by the inhibition of POLA1 or the depletion of the POLA/PRIM complex in DNA replication dynamics we have carried out the following new experiments: first, we have used a combination of 4 siRNA to reduce the levels of the whole POLA/PRIM complex (as opposed to the individual depletion of the subunits); second, we have analyzed the presence of primed DNA in each of the different settings and the effect of VCP/p97 inhibition by the EdU comet assay, as suggested by the reviewer. The new results show different effects of VCPi treatment on replication dynamics when combined with POLAi or POLA/PRIM depletion. Here we summarize the conclusions that we draw from the analysis of DNA replication dynamics, checkpoint activation and priming activity of POLA/PRIM.

The partial inhibition of POLA1: i) increases fork speed and reduces origin firing (Figure 2E-F); ii) induces a modest increase in phosphorylation of CHK1 (Figure 2K); iii) does not change the EdU tail moment (Figure 6A-B). These results are explained by a slow and inefficient priming by POLA/PRIM that leads to a moderate accumulation of ssDNA and activation of ATR, indirectly increasing fork speed.

The complete inhibition of POLA1 blocks DNA replication, strongly reduces fork speed (Reviewer Figure 1), increases the phosphorylation of CHK1 and RPA2 (Figures 2K and 5B) as well as the tail moment for EdU and total DNA (Figure S6C-D). These data agree with the induction of replication catastrophe (Toledo et al., Cell 2013) due to the excessive accumulation of ssDNA. This damage precludes further analysis of the effect of VCP/p97 inhibition under these conditions.

The depletion of the POLA/PRIM complex: i) increases fork speed and reduces origin firing (Figure 4A-B); ii) induces a strong increase in CHK1 phosphorylation (Figure 4C); iii) reduces the EdU tail moment (Figure 6C-D). Compared to the partial inhibition of POLA1, the reduced levels of POLA/PRIM elicit a clear inhibition of priming leading to higher levels of ssDNA and a stronger activation of CHK1 (still lower than the effect of the complete inhibition of POLA1). We propose that fork rate is increased by the reduction in POLA/PRIM since it has been previously shown that the absence of the primase complex allows the faster progression of forks in viral models of replication (Lee et al., Nature 2006). In addition, the activation of CHK1 promotes the activation of the translesion synthesis (TLS) pathway (Li et al., Nat Comm 2024; González-Besteiro et al., EMBO J 2019) that can also promote fork progression.

While the inhibition of VCP/p97 has a very small effect on fork speed after partial POLA1 inhibition (Figure 2E), it reverts the increase in fork speed induced by POLA/PRIM depletion (Figure 4A). We think that POLAi and POLA/PRIM depletion elicit different mechanisms to stimulate fork speed, as explained above. It would be interesting to address these differences in future studies.

Interestingly, VCPi increases the EdU tail moment both in combination with POLAi (Figure 6G-H) and in POLA/PRIM depleted cells (Figure 6I-J). Checkpoint activation depends on the presence of ssDNA and the amount of primed DNA. Thus, the combination of VCPi (increased primed DNA) and partial inhibition of POLA1 (increased ssDNA) leads to a cumulative phosphorylation of CHK1 (Figure 2K). In POLA/PRIM depleted cells the accumulation of ssDNA is reduced by VCPi treatment, while primed DNA is increased by VCPi, resulting in unaltered levels of CHK1 phosphorylation (Figure 4C).

6) According to the authors' model, excess Okazaki fragments serve as recruiting signal for TOPBP1 and consequently ATR/CHK1 activation. As checkpoint activation is expected to affect replication fork progression and origin firing, the authors should carefully discuss what is primary consequence and what is a secondary effect of ATR/CHK1 activation on replication fork properties. **Technically, it would be feasible to combine VCP/POLA inhibition with ATR/CHK1 inhibition to experimentally validate this aspect.**

DNA fiber assays (Figure S5A-B) show that the effects of VCP/p97 on DNA replication dynamics are upstream of ATR/CHK1. The inhibition of CHK1 decreases fork speed and increases origin firing (González-Besteiro *et al.* EMBO J 2019). However, CHK1 inhibition does not revert the changes in fork speed and origin firing induced by VCPi (Figure S5A-B), indicating that the effect of the inhibition of VCP/p97 in replication dynamics is not a consequence of the activation of the RSR.

7) **The representation of DNA fiber patterns in Figure 1C is somewhat confusing. In the upper row, the pulse labelling strategy is shown, while the lower panel depicts the labelling pattern that served as the readout for interpretation. This should be labelled clearly, to avoid confusion of both pictograms. It also remains vague in how far the determined properties of replications forks reflects specific defects in replication regulation, e.g. what does the CldU/IdU ratio report, what is the concept behind fork asymmetry? The authors should define how they interpret different fork properties.**

We have re-organized Figure 1 to better explain how we label the DNA and what we measure in each of the experiments. We have clarified in the manuscript what we measure in the asymmetry experiments.

8) Figure 3B shows an isolation experiment of POLA1 from control and VCP inhibited cell lysates using TUBEs. The fact that the higher molecular weight bands of POLA1 are almost identical in the two input conditions along with the fact that in both conditions mainly unmodified POLA1 was enriched, leaves a doubt about the specificity of a) the TUBE enrichment of ubiquitin conjugates and b) the POLA1 detection. This experiment should be interpreted with caution.

In additional TUBE experiments we were unable to get a clearer result for POLA1. Thus, we decided to use an alternative method to detect POLA/PRIM ubiquitylation. We transfected HCT116 cells with His-tagged ubiquitin and purified the ubiquitylated proteins in cells arrested in S phase in control conditions and after treatment with VCPi. Figure S3C shows that both POLA1 and PRIM2 are ubiquitylated and their modification is increased by the inhibition of VCP/p97. We did not detect the modification of POLA2 or PRIM1.

Reviewer #3 (Remarks to the Author):

The manuscript by Rodriguez-Acebes *et al.*, "The activation of ATR during unperturbed DNA replication is restricted by VCP/p97 through the extraction of DNA polymerase α /Primase from chromatin", aims to address the role of VCP during DNA replication, beyond its known role in termination and prevention of re-replication. Here, data show that VCP inhibition causes several phenotypes: (i) a mild decrease in the progression of forks, and an increase in asymmetric forks (indicative of fork stalling). (ii) A strong decrease in origin firing and reduced EdU incorporation. (iii) A mild but reproducible activation of checkpoint signalling, which causes G2/M arrest, and which can be rescued by inhibition of CHK1 kinase.

Authors suggest that VCP functions in regulating primer synthesis on the lagging strand, following fork activation. The proposed model suggests that VCP removes DNA polymerase α /Primase from chromatin during unchallenged replication, thereby limiting the number of Okazaki fragments and reducing the basal levels of checkpoint signalling. In the absence of VCP, there is over-priming of the DNA, excessive TOPBP1 recruitment and checkpoint signalling. This is an interesting idea, as little is understood about how VCP functions in DNA replication, aside from CDT1 degradation and replisome disassembly, but unfortunately there is currently not enough strong evidence to support the proposed model.

Evidence that VCP and Pol α /PRIM work in the same pathway stems from the fact that inhibition of POLA and VCP, individually or together, causes the same level of fork stalling (FIG 2G), decreased origin firing (FIG 2F) and both prevent the increase in fork speeds caused by CDC7i. The checkpoint signalling seen in the VCPi-treated cells is also certainly replication-dependent, because co-inhibition of CDC7 (FIG 3D) and co-depletion of TOPBP1 (FIG S3F), which are known to promote fork activation, lead to convincing reductions in CHK1 phosphorylation.

We thank the reviewer for the kind words about our work and its relevance.

Many elements of the proposed model are however not supported strongly enough by the experimental data. This is especially the case for the direct role of VCP removal of POLA/primase from chromatin.

We have improved the data regarding the interaction of VCP/p97 with the POLA/PRIM complex and added new experiments confirming that POLA/PRIM accumulates on chromatin upon VCP/p97 inhibition using complementary approaches (detailed below).

While this work is interesting and worthwhile investigating, it requires more work to support the proposed model before being ready for publication in Nature Communications.

Major comments

1. The molecular weight markers are missing on all blots throughout the paper and it is difficult to judge which of the multiple bands is of a correct size and whether ubiquitylated forms are mono or polyubiquitylated. It is of importance as VCP shows a threshold of 5 ubiquitins added for processing.

We have added the markers to all the Western blots.

2. Many of the accumulation or checkpoint activation effects observed by western blotting are rather mild and raise questions of reproducibility and robustness. Quantification of a number of experiments would increase the confidence in the proposed conclusion.

We have quantified the CHK1-P/CHK1 ratio in the following experiments:

1. Effect of VCP/p97 inhibition (Figure 1I quantified in Figure S1E, 4 replicates)
2. Combined effect of VCP/p97 and POLA1 inhibition, dose dependency (Figure 2K quantified in Figure S2G, 2 replicates)
3. Effect of VCP/p97, POLA1 and CDC7 inhibitors (Figure 3F quantified in Figure S3F, 7 replicates)
4. Effect of VCP/p97 inhibition in POLA/PRIM depleted cells (Figure 4C-D, 5 replicates)

For the accumulation of the POLA/PRIM complex on chromatin, we have carried out immunofluorescence analysis after pre-extraction of the nuclear soluble material. Figure 3D-E shows that both POLA1 and PRIM2 accumulate on chromatin upon inhibition of VCP/p97.

3. Authors need to address whether any of the VCPi phenotypes they observe could be explained by the known role of VCP in degrading CDT1 and the subsequent re-replication that would ensue. This role is mentioned in introduction but ignored afterwards.

Recent data shows that CDT1 is stable for 30 min after entry into S phase and then degradation takes place leading to DNA synthesis (Ratnayake et al., Mol Cell 2023). Our experimental conditions involve the synchronization of cells in G1/S and release for 2 h before the treatment with VCP/p97 inhibitor, at a point where DNA replication is at its maximum. At this stage the remaining pool of CDT1 should no longer be under the control of VCP/p97. In fact, our results show that CDT1 is not significantly stabilized on chromatin even after 6 h of treatment with VCPi (Reviewer Figure 2).

4. Figure 3B lacks necessary controls – currently it looks like there is more ubiquitylated POLA in input than in TUBE pulldown. Is there enrichment of any ubiquitylated products in TUBE pulldown over nonspecific beads? Is there enrichment of ubiquitylated POLA in TUBE pulldown over nonspecific beads? Is this reproducible?

Since the TUBE experiments were leading to dirty results with many non-specific bands, we decided to use an alternative method to analyze POLA/PRIM ubiquitylation. To this end we purified ubiquitylated proteins from

HCT116 cells after the overexpression of His-tagged ubiquitin. Figure S3C now shows that POLA1 and PRIM2 are ubiquitylated and their modification is increased after the inhibition of VCP/p97. We have not detected ubiquitylation of POLA2 and PRIM1.

If VCP is required for continuous removal of excess POLA/primase from unchallenged replication forks then one can expect this interaction to be more robust (especially after inhibition of p97 activity), but very little interaction is observed (Fig 3 and S3). Moreover, one would expect interaction of VCP with polyubiquitylated forms of POLA, while authors focus on non-modified POLA. Similarly, one would expect accumulation of ubiquitylated forms of POLA on chromatin (and figure 3B suggest that these can be visible in input), but only unmodified forms are presented in Figure 3C.

We have improved the immunoprecipitation data in the manuscript, focusing on the pull-down of UFD1L and PRIM2. Figure 3A-B shows a more robust interaction between VCP/UFD1L and the POLA/PRIM complex. Conversely, we have detected POLA1 and PRIM2 in the material pulled-down with UFD1L. In addition, Figure S3B shows that the depletion of POLA1 reduces the interaction of UFD1L with the whole complex as we observe reduced amounts of PRIM2 in the immunoprecipitation of UFD1L even if the levels of PRIM2 are not changed upon downregulation of POLA1. These data suggests that POLA1 is responsible, at least in part, for the interaction of the POLA/PRIM complex with UFD1L. Since we detect the accumulation of POLA1 on chromatin induced by VCPi by IF (Figure 3D-E) but we do not see this accumulation by Western blot (Figure 3C), we conclude that ubiquitylated POLA1 cannot be detected by Western blot under these conditions. Instead, His-ubiquitin experiments show that ubiquitylated POLA1 migrates as a high molecular weight smear (Figure S3C). Thus, we do not expect to detect ubiquitylated POLA1 in pull-down experiments either.

Moreover, the conclusion that VCPi does not affect retainment of Cdc45 and PCNA on chromatin is misleading as the provided timecourse does not cover whole replication reaction – the level of PCNA and Cdc45 does not change over the timecourse, suggesting that cells do not reach end of the S-phase.

The analysis of the cell cycle by flow cytometry confirms that we do not reach the end of S phase in our experimental conditions (Reviewer Figure 3) and we have changed the text accordingly. However, this does not change the conclusions of our work.

Reviewer Figure 3. Cell cycle progression in synchronized HCT116 cells treated with VCPi.

HCT116 cells were synchronized in G1/S in the presence of 2 mM thymidine for 16 h. Cells were released for 2 h and then treated with DMSO (Control) or 5 μ M NMS875 (VCPi) for the indicated times. The cell cycle progression was measured by flow cytometry, following total DNA content with DAPI and DNA replication with EdU labelling.

5. TOPBP1 is needed for checkpoint activation but also for initiation of DNA replication. Upon downregulation of TOPBP1 (Fig S3F) it is likely that observed effects are due to inhibition of origin firing (akin CDC7i) rather than loading of TOPBP1 on primers in order to activate checkpoint. There is many controls required to draw the conclusion proposed.

As the reviewer points out TOPBP1 is necessary for DNA replication initiation and is therefore essential at the cell level. However, an almost complete depletion of TOPBP1 is necessary to affect cell proliferation, as shown in HCT116 cells expressing a degron tagged TOPBP1 (Bass and Cortez, JCB 2019). This work shows that a small amount of TOPBP1 still sustains DNA replication. In our experimental setting cells that are still proliferating should have enough TOPBP1 to support DNA replication initiation but fail to activate ATR. Further, the reduction in CHK1 phosphorylation induced by TOPBP1 depletion (Figure S3G) is much stronger than the effect observed upon CDC7 inhibition (Figure 3F). While we cannot exclude a small effect of DNA replication initiation inhibition by TOPBP1 depletion this effect should be minor.

6. The presented experiments with downregulation of different components of POLA/primase complex are very confusing as showing very different phenotypes for downregulation of different subunits. If the major function of these subunits is to fulfil POLA/primase role as a part of the complex, one would expect them to have similar phenotypes. As it stands these differences suggest off-target effects of different siRNAs used, and suggest that

authors should rescue the observed phenotypes using siRNA-resistant re-expression of proteins to confirm that these are not off-target artefacts.

We believe that the differences in phenotype observed with the individual downregulation of the POLA/PRIM subunits stem from the degree of depletion achieved for each subunit of the complex with each siRNA. To circumvent this issue, we have combined the 4 individual siRNA at a lower concentration to achieve a depletion of the whole complex. At this concentration, the off-target effects of the individual siRNA should be negligible. As shown in Figure S4A, this strategy leads to an efficient depletion of POLA/PRIM, except for a less pronounced effect on POLA2 levels. We have repeated the key experiments of the manuscript upon downregulation of the complex (Figure 4) and we have added new experiments using the EdU comet assay (Figures 5-6).

As expected, the depletion of the POLA/PRIM complex increased fork rate and reduced origin firing (Figure 4A-B). It led to a strong activation of CHK1 (Figure 4C) most likely due to the accumulation of ssDNA (reflected by phosphorylated RPA2, Figure 4C). The depletion of POLA/PRIM induces problems during fork progression and increased fork asymmetry (Figure 4E-F). The EdU comet assay confirmed the reduction in priming in POLA/PRIM depleted cells (Figure 6C-D).

In POLA/PRIM depleted cells VCPi treatment reduced fork speed (Figure 4A) and the levels of ssDNA as observed by a reduction in RPA2 phosphorylation (Figure 4C), while it still increased priming (Figure 6I-J). The reduction in ssDNA compensates the accumulation of primed DNA resulting in unchanged levels of CHK1 phosphorylation (Figure 4C).

7. If VCPi reduces priming while also reducing general origin firing and DNA synthesis, one would expect not more Okazaki fragments after VCPi, but shorter Okazaki fragments. Authors should use alkaline gel electrophoresis to measure the size of Okazaki fragments more precisely and also compare VCPi-treated cells with POLAi-treated cells and be able to rescue the phenotype.

Our main objective was to show that the control of priming by VCP/p97 is important for the activation of CHK1 in an unperturbed DNA replication. To this end we have expanded our experiments with the EdU comet assay (Figures 6 and S6), confirming that VCP/p97 controls the priming action of POLA/PRIM and that the activation of the RSR is regulated by this action (Figure 5). Setting up the alkaline gel electrophoresis would involve long time and a strong effort. We think that measuring the length of the Okazaki fragments will not provide further proof that VCP/p97 controls POLA/PRIM and does not contribute to the central message of the manuscript.

8. Finally, in order to confirm this model further, authors should identify the VCP cofactor(s) implicated in this mechanism as this would help to relieve some of the pleiotropic effects of VCP inhibition and make the data much more robust.

We have carried out some preliminary experiments to explore the VCP/p97 cofactors that mediate the interaction with POLA/PRIM. We show in the manuscript that UFD1L interacts with POLA/PRIM (Figure 3A-B). The UFD1L-NPLOC4 dimer has been involved in the action of VCP/p97 on its substrates in chromatin. Additional co-factors associate with the VCP-UFD1L-NPLOC4 complex for the recognition of substrates. Our preliminary results indicate that the depletion of UFD1L is not sufficient to break the interaction of VCP/p97 and POLA/PRIM (Reviewer Figure 4) thus pointing to the involvement of additional cofactors. We are currently exploring the identity of these cofactors, which will be the subject of follow-up studies.

Reviewer Figure 4. Immunoprecipitation of PRIM2 in UFD1L depleted cells.

HCT116 cells were transfected with a control siRNA (siC) or an siRNA directed against UFD1L (siUF). 24 hrs after transfection cells were synchronized in the presence of 2 mM thymidine for 16 hrs. Cells were released for 2 hrs and then treated with DMSO (C) or 5 mM NMS873 (VCPi). PRIM2 was immunoprecipitated from total nuclear extract (PRIM2) and a non-specific IgG was used as a control (IgG). The presence of POLA1, VCP/p97, PRIM2, UFD1L and histone H2A was followed by Western blot with specific antibodies. 2% of the input is shown.

Minor comments

1. What are the units the tail moment is quantified in Figure 5F and Figure S6?

The tail moment is a relative measure of the fluorescence intensity in the tail compared to the intensity in the head of the comet and thus it is expressed in arbitrary units. We have corrected the figures accordingly.

2. It would be helpful to explain at the beginning of the study the mode of action of POLAi and concentration required to inhibit its activity. Only towards the end of the manuscript it is revealed that the concentration used throughout does not strongly block POLA activity. The direct measure of the level of inhibition would also help to support the conclusions drawn.

We have changed the manuscript to make clear earlier on that we are using only a partial inhibition of POLA1 activity in most of the experiments. The level of inhibition has been thoroughly analyzed in previous works (Abdel-Samad et al., Am J Cancer Res 2018) and nicely correlates with the accumulation of ssDNA and a limited amount of DNA damage (Ercilla et al., Cell Rep 2020 and this manuscript). We hope that the changes in the manuscript have clarified the conclusions.

Reviewer #1 (Remarks to the Author):

The authors have carefully addressed all my points and have significantly improved the manuscript.

1. The new interaction data in Fig 3A (together with knockdown control in Fig S3B) is a big improvement and strengthens the conclusions considerably.
2. The His-Ubiquitin approach in Fig S3C is also much better than the previous TUBEs approach. If the authors have (negative) data on POLA2 and PRIM1 using this approach, it would be worthwhile to include this too.
3. The new POLA1 imaging data in Fig 3D, and the combined POL/PRIM knockdown in Fig 4C/Fig S4A is much more convincing now.

Reviewer #2 (Remarks to the Author):

The authors have convincingly addressed all the issues raised.
The manuscript is a strong contribution to the broad readership of Nature comms.

Authors' response: We thank both reviewers for their help in improving the manuscript.

Reviewer #3 (Remarks to the Author):

The revised version of the manuscript by Rodriguez-Acebes et al. has improved and thank you for addressing a number of my and other reviewers comments. Thank you for the markers, quantifications etc.

I still believe that this is an interesting problem, but still have a number of questions about the key experimental findings of the manuscript.

The authors removed some of the problematic experiments and provided new ones, but the new ones lead to new questions and are missing controls.

Thank you for appreciating the improvements in our manuscript. We have carried out additional controls as requested and we also provide additional data to further support our conclusions. We understand that some of the new results may suggest new questions but we think that the manuscript already provides a significant advance of broad interest in the field, as indicated by the other reviewers. The detailed mechanistic aspects of the extraction of POLA/PRIM from chromatin and the study of how POLA1 inhibition affects DNA replication are current and future projects in the lab. We include in this letter some of the results we have obtained in these areas for the reviewer's perusal.

The main part that is not convincing is the data supporting the main hypothesis that VCP/p97 removes ubiquitylated POL1/PRIM from chromatin.

This is based on:

1. - interaction of POL1/PRIM2 with VCP/UFD1 – this is now mainly based on UFD1 immunoprecipitation (Figure 3B), however in this IP UFD1 does not interact with VCP/p97? One would expect UFD1 to interact robustly with VCP/p97 – is this then a subcomplex that does not work with VCP/p97?

The interaction of VCP/p97 with its adaptors is very dynamic and a robust interaction can only be detected after crosslinking, as shown by the Deshaies group (Xue et al., 2016 Mol Cell Proteomics). Since the pull-down of POLA/PRIM was not efficient in these conditions we decided to stick to immunoprecipitation in the absence of crosslinking. Under these conditions we consistently recover detectable amounts of VCP/p97 in the UFD1L pull-down. We include here an additional immunoprecipitation experiment in which we have loaded a lower amount of the input to improve the visualization of the pull-down of VCP/p97 with UFD1L. These results are consistent with the current model of action of VCP/p97 and its adaptors.

Reviewer Figure 1. Western blot analysis of the immunoprecipitation of PRIM2 (left) and UFD1L (right) from HCT116 whole nuclear extracts.

HCT116 cells were synchronized at G1/S, released for 2.5 h and treated for 3 h with 5 μ M NMS873 (VCPi) or DMSO as a control (C). 0.5% of the input material is shown and a non-specific IgG was used as a negative control. The levels of POLA1, PRIM2, VCP/p97 and UFD1L were analyzed with specific antibodies.

2. - retention of POLA1/PRIM2 on chromatin in VCPi – this is not visible on chromatin by western blot and authors argue that this is due to inability to visualise ubiquitylated forms of these proteins – what about treating samples with DUBs that would cut off the chains and visualise accumulated unmodified proteins?

We would like to point out that an increased retention of PRIM2 is visible by Western blot as shown in Figure 3C, even after only 30 minutes of incubation with VCPi. As suggested by the reviewer, we tried to incubate the chromatin fractions obtained from DMSO- or VCPi-treated HCT116 cells with the USP2 catalytic domain to remove protein ubiquitylation before the analysis by Western blot. However, we were not able to achieve an efficient removal of protein ubiquitylation under our experimental conditions. We have added in this letter two additional pieces of data in support of our explanation about the lack of accumulation of POLA1 on chromatin by Western blot after VCPi treatment. First, we have checked the specificity of the immunofluorescence of POLA1 (see below), confirming that the levels of POLA1 on chromatin increase in response to VCPi. In addition, we have shown that we can induce the accumulation of

POLA/PRIM on chromatin through the depletion of specific VCP/p97 cofactors (see below). In line with the results using VCPi, the levels of POLA2, PRIM1 and PRIM2 on chromatin are clearly increased while the changes in POLA1 are only minor.

Second way of visualising POLA1/PRIM2 on chromatin that authors provided now is IF. However, the experiments provided have no controls. Could the authors provide negative controls for specificity of antibodies – e.g. siRNA of PRIM2/POLA1 to show that signal disappears, look at signal at different cell cycle stages to get a base line in G1, increase in S-phase and then even more accumulation in VCPi, positive control with some known treatment that increases POL alpha on chromatin. This is a key experiment and needs controls.

We agree with the reviewer that it was necessary to control for the specificity of the antibodies in the immunofluorescence (IF) assays. To this aim, we have repeated the IF stainings after siRNA-mediated downregulation of POLA1 or PRIM2. As shown in Reviewer Figure 2, POLA1 IF staining is fully validated. The fluctuations of POLA/PRIM complex on chromatin in the cell cycle are an interesting but complementary question that is part of our follow-up project to study the mechanisms of extraction of POLA/PRIM by VCP/p97.

On the other hand, the IF staining of PRIM2 was not abolished by siRNA. Therefore, we have opted to remove PRIM2 IF data from the manuscript and we thank the reviewer for bringing up this important issue.

Reviewer Figure 2. Control of POLA1 immunofluorescence.

Immunofluorescence of HCT116 cells transfected with an siRNA against POLA1 (siPoA1) or a non-specific siRNA as a control (siC). Nuclei were stained with DAPI. The quantification of the experiment is shown in the right.

3. – accumulation of ubiquitylated forms of PRIM2/POLA1 in VCPi treatment – authors here exchange tube experiment for His-Ubi pulldown, but the new experiment although showing showing robust ubiquitin pulldown, presents very minimal levels of POLA1/PRIM2 signal.

There are several reasons that could explain the limited signal strength for ubiquitylated POLA1 and PRIM2 in Figure S3C. First, we only expect a fraction of the POLA/PRIM complex to be modified since its ubiquitylation is restricted to chromatin, and most likely only a fraction of POLA/PRIM on chromatin is ubiquitylated. In addition,

it is hard to assess the fraction of ubiquitylated protein from a Western blot since the signal of the smear of the modified protein is not directly comparable to the signal of a single band. Last, the antibodies against POLA1 and PRIM2 are not very clean in Western blot, making the detection of the modified protein more challenging. We have repeated the purification of ubiquitylated proteins and we have gotten a better signal for PRIM2 ubiquitylation (included in new Figure S3C), which is very consistent throughout the experiments.

Altogether, with the three main three lines of evidence having problems, I am still not convinced that POLA1/PRIM2 is a VCP target.

We have strengthened the three lines of evidence with the following data:

1. An additional immunoprecipitation experiment showing the pull-down of VCP/p97 by UFD1L.
2. The confirmation of the specificity of the POLA1 signal in immunofluorescence.
3. A better immunoblot showing the ubiquitylation of PRIM2 protein.

In the previous rebuttal letter we had included data showing that depleting UFD1L did not affect the interaction of PRIM2 and VCP/p97. These results are part of an ongoing follow-up project in the lab where we intend to characterize the mechanism of extraction of POLA/PRIM by VCP/p97 in detail. We have included additional data on this point for the reviewer's information. We show that POLA/PRIM accumulates on chromatin after the depletion of FAF2 or NPLOC4 (which also reduces the levels of UFD1L; Reviewer Figure 3). Conversely, we observe a reduction in the amount of POLA/PRIM in the soluble nuclear fraction (Reviewer Figure 3). These data further support that VCP/p97 extracts POLA/PRIM from chromatin in S phase with the help of specific cofactors as part of the new project currently under development in the lab.

Reviewer Figure 3. Chromatin localization of the POLA/PRIM complex upon depletion of FAF2 and NPLOC4. HCT116 cells were transfected with an siRNA against FAF2 or NPLOC4, using a non-specific siRNA as control (C). After synchronization in S phase, the chromatin and the soluble nuclear fractions were obtained and analyzed by Western blot with the indicated antibodies.

I appreciate also that alkaline gel electrophoresis is a new technique to be set up, however I believe that it provides much more information than tail moment analyses provided. Tail moment measured can be effect of any fragments of DNA containing

nascent DNA and does not allow to distinguish the differences in the length and abundance of the Okazaki fragments.

We respectfully disagree on this point. The alkaline gel electrophoresis detects Okazaki fragments as well as other fragments containing nascent DNA, similar to the EdU comet assay. Even if we cannot determine the size of these fragments in the EdU comet assay, we can measure their abundance through the tail moment. We have carried out additional experiments with the EdU comet assay to further validate the technique and to get a deeper understanding of the effect of POLA1 inhibition (see below).

To convincingly show that VCPi affects priming more detailed analysis is needed.

The EdU comet assay can monitor the maturation of nascent DNA fragments by performing pulse-chase experiments that follow the progressive loss of the EdU tail (Stoy et al., NSMB 2023). As shown in Reviewer Figure 4, the treatment with VCPi led to increased EdU tail moment at all chase times, in line with our previous results, and in agreement with increased priming by POLA/PRIM. We have also verified that control and VCPi-treated cells show a similar rate of reduction of EdU tail moment (Reviewer Figure 4). As the reviewer previously pointed out, it could be anticipated that increased priming would result in shorter Okazaki fragments that would mature at a faster rate, but it should be taken into account that VCP/p97 inhibition results in slower forks (Figure 1D, 2A,C,E). Both effects could compensate each other, explaining the similar rate of EdU tail moment decrease.

Reviewer Figure 4. Okazaki fragment maturation analysis in HCT116 cells.

Comet assay performed in HCT116 cells synchronized by a single thymidine block and released for 2 h. Nascent DNA was labelled with a pulse of 30 μ M EdU for 45 min and then chased for the indicated time in the presence of 5 μ M NMS873 (VCPi) or DMSO as control. The experimental setup is shown in top. The EdU was conjugated to a fluorescent probe by a Click reaction after the alkaline comet assay. The left panel shows the EdU tail moment from 4

independent experiments; the central panel shows the slope of the reduction in EdU tail moment in control and VCPi treated cells; the right panel shows the normalized value of the median of the 4 independent experiments compared to the 1 h time point.

In addition, we draw the following conclusions from these experiments:

1. VCP/p97 inhibition does not affect the process of Okazaki fragment maturation.
2. VCP/p97 inhibition does not induce DNA damage in these conditions. As shown by Stoy et al. (NSMB 2023), the induction of DNA damage delays the disappearance of the EdU tail. The dynamics of maturation in the presence of VCPi are not significantly altered, in line with our results in Figure 1I that showed no induction of DNA damage after 30 min of VCPi treatment.

These figures may become part of a Supplementary figure if the reviewer or Editor consider it appropriate.

For example – the low concentration of POLAi induces inhibition of new origin firing and change of fork speed (Figure 2) but not changes in tail moment, interpreted as no changes in priming (Figure 6). Where does the effect observed in fibres come from then, if not from differences in priming?

Our manuscript focuses on the functions of VCP/p97 in the control of DNA replication and the replication stress response through the extraction of POLA/PRIM, rather than how DNA replication responds to POLA1 inhibition. We appreciate that this is a very interesting point and we are planning to address it in the future.

In any case, to further substantiate our finding that partial POLA1 inhibition does not affect priming, we have analyzed the effect of increasing doses of POLAi (adarnotene) in priming and DNA damage. Reviewer Figure 5 shows that there is no significant change in EdU tail moment or total DNA tail moment at 0.2 or 0.5 μ M POLAi (left and center). At 0.5 μ M POLAi, a moderate reduction in total EdU signal intensity is detected (right) even if there are no changes in EdU tail moment and no induction of DNA damage. At higher concentrations of POLAi (1 μ M and above) we observed a very strong reduction in total EdU incorporation along with a strong increase in both total DNA and EdU tail moment. Thus, the complete inhibition of POLA1 leads to a block in DNA replication and the appearance of DNA damage, while partial block of POLA1 activity does not change priming or induce DNA damage. These results are in line with the previous observations by Ercilla *et al.* (Cell Reports 2020), the data presented in our manuscript (Figures 6A-B, S6A, S6C-D), and the distributive mode of action proposed for POLA/PRIM (Porcella *et al.*, PLoS Genet 2020). We could add these data in the manuscript should the reviewer or editor request it.

Reviewer Figure 5. Comet assay on nascent DNA labelled with EdU.

Cells were incubated with 30 μM EdU for 60 min, and EdU was conjugated to a fluorescent probe by a click reaction after alkaline comet assay. HCT116 cells were synchronized, released for 2.5 h and treated for 2 h with increasing concentrations of adarotene (POLAi) or DMSO as a control. The EdU tail moment (left), total DNA tail moment (center, SYBR) and total EdU intensity (right) were measured. Three independent experiments were combined and the median of individual experiments is shown with different symbols.

We hypothesize that the accelerated fork rate upon partial inhibition of POLA1 is a consequence of impaired origin firing that activates compensatory mechanisms to promote full replication of the DNA. Our preliminary data, provided here for the reviewer's perusal, suggests that the increase in fork speed induced by POLA1 inhibition is partially prevented in cells deficient for PRIMPOL (Reviewer Figure 6) and strongly prevented by the inhibition of Rev1 that blocks the translesion synthesis pathway (Reviewer Figure 7).

Reviewer Figure 6. DNA fiber analysis in HCT116 cells, wild type (WT) and PrimPol KO. Cells were synchronized, released for 2 h and treated for 2 h with 0.5 μM POLAi or DMSO as a control. Fork rate (left) and 1st label origins (right) were determined. Two independent experiments were pooled and the individual medians are shown.

Reviewer Figure 7. DNA fiber analysis in HCT116 cells, synchronized, released for 2 h and treated for 2 h with 0.5 μ M POLAi, 1.5 μ M TLSi (JH-RE-06), a combination of both or DMSO as a control. Fork rate (left) and 1st label origins (right) were determined. Two independent experiments were pooled and the individual medians are shown.

Finally, two of the reviewers highlighted the pleiotropic effects of VCPi and the need to narrow down the cofactors responsible, to have a parallel path of validation of the mechanism beyond the pleiotropic VCP. There is a handful of known cofactors working with VCP on chromatin and it is not beyond the scope to siRNA them and see which one reproduces the effects on POL alpha observed with VCPi.

Indeed, we are interested in the identification of the relevant cofactors for the extraction of POLA/PRIM by VCP/p97. However, even if the number of known cofactors is not that high, their combinatorial mode of action complicates their analysis. We think that we have convincingly shown that POLA/PRIM is a substrate of VCP/p97 and including the preliminary data on the cofactors responsible for this action (Reviewer Figure 3) will not add any significant conclusion in the manuscript. As discussed above, new results open new interesting questions and we plan to tackle them in the future. Similarly, answering how the partial inhibition of POLA1 leads to increased replication fork speed is a very interesting new project that goes beyond the focus of this manuscript.

Reviewer #3 (Remarks to the Author):

I would like to thank the authors for extensive response to my comments in the authors' rebuttal. I am a little disappointed that not much of what is shown in the rebuttal has made it into the manuscript itself.

We want to thank the reviewer for the useful comments on our manuscript. We included most of these data in the rebuttal to not overload the manuscript. We will move some of these results to the final version of the manuscript.

Thank you for showing me that the immunofluorescent antibody against PolA1 is specific, however, I believe that this is an essential control that should be included within the manuscript, too. Thank you also for removing the staining with PRIM2 that turned out not to be specific.

We have included the control for POLA1 antibody specificity in Supplementary Figure 3D.

Thank you for the new version of the HIS-Ubi pulldown – indeed, it does look better.

Thanks for helping us improve the quality of our work.

The authors showed just in rebuttal a different version of the PRIM2 and UFD1L immunoprecipitation, that indeed shows some VCP interacting with UFD1L. I can see how the authors may not wish to repeat the same blot within manuscript, but maybe they could include quantification of the IP data to indicate reproducibility of their results over a number of experiments?

In new Supplementary Figure 3B-C, we have quantified the amount of POLA1, VCP and UFD1L in four immunoprecipitation experiments of PRIM2, and the amount of POLA1, VCP and PRIM2 in four immunoprecipitation experiments of UFD1L. We have normalized the levels of each protein in the pull-down to the levels of protein in the input, showing the non-specific IgG pull-down as a control. We have only shown the quantification in control conditions.

The quantification of these blots presents some caveats, as the background in each lane strongly differs. In general, the IgG lanes present a higher background which explains why the quantification shows relatively high levels of POLA1 and PRIM2 in the IgG control, which are higher than what is actually observed in the blots. Even though, the quantification still shows a clear enrichment for all the proteins in both PRIM2 and UFD1L immunoprecipitations.

The authors included also in rebuttal data on downregulation of one of the major cofactors of VCP (NPLOC4) and one of the minor cofactors (FAF2), which lead to small accumulation of POLA/PRIM. These data does strengthen the main message of this manuscript – that VCP controls the level of chromatin bound POLA/PRIM, and also addresses the point raised by two of the reviewers about the pleiotropic nature of inhibition of all of the VCP activity when using the inhibitor. I strongly believe that these

data should be included in the manuscript to strengthen it. The authors present these data as a response to 3 of my points, but do not include them in the manuscript.

We understand the point of the reviewer. However, we think that these data are not easy to integrate in the manuscript. These adaptors would be only mentioned in this experiment and there would be no justification as to why these adaptors, and not others, are specifically analyzed. Further, we think that these data would be better suited for the analysis of the mechanisms of extraction of POLA/PRIM by VCP/p97, which we are currently working on. Thus, we have opted to leave this data out of the manuscript.

Finally, my understanding of the description of the chosen low concentration of POLAI inhibitor, here and in cited previous papers, is that the moderate accumulation of ssDNA and RPA on chromatin, and inhibition of origin firing, results from low level of inhibition of priming activity provided by POLA/PRIM. However, if the authors believe that there is no effect on priming, based on their EdU and DNA tail moment (manuscript and rebuttal), then this is an important message to introduce as a concept when introducing the experimental setting on page 7. This will disperse the assumption that POLAI inhibitor inhibits the priming activity at all concentrations that show phenotypic changes.

As suggested by the reviewer, we have now included the data on the dose dependent effect of POLA1 inhibition in Supplementary Figure 2B-D. We have changed the text accordingly, explaining early in the manuscript how our data indicate that partial inhibition of POLA1 does not affect priming.

Overall, I believe that the indicated above information that was provided just for reviewers is important to include within the manuscript.

Thank you for addressing my comments.

Again, we thank the reviewer for helping us improve our work. As requested, we have included most of the data from the rebuttal to clarify some of the points of the manuscript.